



# An adaptation of the $CO_2$ slicing technique for the Infrared Atmospheric Sounding Interferometer to obtain the height of tropospheric volcanic ash clouds

Isabelle A. Taylor[1,2], Elisa Carboni[2], Lucy J. Ventress[3], Tamsin A. Mather[1], and Roy G. Grainger[2]

[1]COMET, Department of Earth Sciences, University of Oxford, Oxford, OX1 3AN, UK
[2]COMET, Sub-Department of Atmospheric, Oceanic and Planetary Physics, University of Oxford, Oxford, OX1 3PU, UK
[3]NCEO, Sub-Department of Atmospheric, Oceanic and Planetary Physics, University of Oxford, Oxford, OX1 3PU, UK (now at STFC RAL Space, Harwell, Didcot, OX11 0QX)

**Correspondence:** Isabelle Taylor (isabelle.taylor@earth.ox.ac.uk)

**Abstract.** Ash clouds are a geographically far reaching hazard associated with volcanic eruptions. To minimise the risk that these pose to aircraft and to limit disruption to the aviation industry, it is important to closely monitor the emission and atmospheric dispersion of these plumes. The altitude of the plume is an important consideration and is an essential input into many models of ash cloud propagation. $CO_2$ slicing is an established technique for obtaining the top height of meteorological clouds

5 and previous studies have demonstrated that there is potential for this method to be used for volcanic ash. In this study, the $CO_2$ slicing technique has been adapted for volcanic ash and applied to spectra obtained from the Infrared Atmospheric Sounding Interferometer (IASI). Simulated ash spectra are first used to select the most appropriate channels and then demonstrate that the technique has merit for determining the altitude of the ash. These results indicate a strong match between the true heights and $CO_2$ slicing output with a root mean square error (RMSE) of less than 800 m. Following this, the technique was applied to

10 spectra obtained with IASI during the Eyjafjallajökull and Grimsvötn eruptions in 2010 and 2011 respectively, both of which emitted ash clouds into the troposphere, and which have been extensively studied with satellite imagery. The $CO_2$ slicing results were compared against those from an optimal estimation scheme, also developed for IASI, and a satellite borne LiDAR is used for validation. Overall, the $CO_2$ slicing tool performs better than the optimal estimation scheme. The $CO_2$ slicing heights returned a RMSE value of 2.2 km when compared against the LiDAR. This is lower than the RMSE for the optimal estimation

15 scheme (2.8 km). The $CO_2$ slicing technique is a relatively fast tool and the results suggest that this method could be used to get a first approximation of the ash cloud height, potentially for use for hazard mitigation, or as an input for other retrieval techniques or models of ash cloud propagation.

*Copyright statement.* TEXT





## 1 Introduction

Encounters of aircraft with volcanic ash have demonstrated that such occurrences can cause significant damage to the plane (Casadevall, 1994; Dunn and Wade, 1994; Pieri et al., 2002; Guffanti and Tupper, 2015). In extreme cases, these have resulted in engine failure (Miller and Casadevall, 2000; Chen and Zhao, 2015) and subsequently life-threatening circumstances. Ash

clouds are closely monitored by the Volcanic Ash Advisory Centres (VAACs) who use a variety of data sources including information from volcano observatories and satellite data (Prata and Tupper, 2009; Thomas and Watson, 2010; Lechner et al., 2017). This allows informed decisions on the closure of airspace following an eruption, which can result is severe disruption and have significant financial implications. For example, the eruption of Eyjafjallajökull in 2010, resulted in the closure of a large portion of Northern European airspace and subsequently, the cancellation of 100,000 flights and a revenue loss of

$1.7 billion (IATA Economic Breifing, 2010). Alongside these potential impacts to the aviation industry, volcanic ash is also a hazard to health (Horwell and Baxter, 2006; Horwell, 2007) and can cause considerable damage to infrastructure (Durant et al., 2010; Wilson et al., 2012, 2015).

Satellite remote sensing, particularly infrared instruments, has been widely used for monitoring the hazards presented by volcanic ash. This has included detection schemes which flag pixels that contain volcanic ash (e.g. Prata, 1989a, b; Ellrod et al.,

2003; Pergola et al., 2004; Filizzola et al., 2007; Clarisse et al., 2010; Mackie and Watson, 2014; Taylor et al., 2015). Other methods have been developed to quantify parameters such as the mass, ash optical depth (AOD), effective radius and altitude of the ash cloud, usually relying on look up tables or optimal estimation techniques (e.g. Wen and Rose, 1994; Yu et al., 2002; Watson et al., 2004; Corradini et al., 2008; Gangale et al., 2010; Francis et al., 2012; Grainger et al., 2013; Pavolonis et al., 2013).

Knowing the position of the ash cloud in three dimensions is critical for hazard mitigation. Plume height is a crucial part of this and it is also a variable in models of ash cloud propagation (Mastin et al., 2009; Stohl et al., 2011; Bonadonna et al., 2012) such as HYSPLIT (Draxier and Hess, 1998; Stein et al., 2015) or NAME (Jones, 2004; Witham et al., 2012). A number of different methods have been used to obtain the height of volcanic ash clouds. These have included the use of ground based and airborne instruments, and satellite techniques (Glaze et al., 1999), some of which are summarised in table 1.

This problem is not unique to volcanic ash. Similar retrieval techniques exist to obtain the cloud height of meteorological clouds. One such method, known as the $CO_2$ slicing technique, described in more detail in section 2, has been widely used to obtain the cloud top height and has been adapted for numerous instruments, as illustrated in table 2. The method has been shown to have some potential when applied to volcanic ash using the Moderate Resolution Imaging Spectroradiometer (MODIS) (Richards, 2006; Tupper et al., 2007). In this study, the technique has been adapted for the Infrared Atmospheric

Sounding Interferometer (IASI; see section 3) and applied to volcanic ash. It was first applied to simulated ash spectra (section 4) to select the most appropriate channels and to demonstrate that the method has promise when applied to volcanic ash. Following this it was applied to scenes containing volcanic ash from the Eyjafjallajökull and Grimsvötn eruptions (section 5) where it was compared against an existing method for obtaining the height of volcanic ash and data from a satellite borne





LiDAR. The results indicate that this method could be applied to get a first approximation of the ash cloud height which could then be used for hazard mitigation and as a parameter in other retrieval methods or ash models.

## 2  CO$_2$ Slicing

The CO$_2$ slicing technique is an established method, developed for obtaining the cloud top height/pressure of meteorological cloud (Chahine, 1974; Smith and Platt, 1978; Menzel et al., 1983). Over the past four decades this tool has been adapted for different instruments, summarised in table 2, including both airborne and satellite platforms. The technique uses a CO$_2$ absorption feature within the thermal infrared part of the electromagnetic spectrum between 665 and 750 cm$^{-1}$ (13.3 to 15 $\mu$m). Within this region, as wavenumber increases there is a general increase in the radiance observed. This is demonstrated in Fig. 1a which shows the spectrum of a simulated clear atmosphere. This has been simulated with the fast radiative transfer model RTTOV and replicates what would be observed with IASI given specified atmospheric conditions. In this case a default atmospheric profile is used, without the addition of cloud, volcanic ash or any trace gases or aerosols above background levels.

Assuming an atmosphere which is decreasing in temperature with height, the radiances measured by the instrument are proportional to the transparency of the atmosphere for each channel (Holz et al., 2006). Subsequently, within the CO$_2$ absorption band, as wavenumber and the radiance measured both increase, the channels are becoming increasingly transparent (with some fluctuations). As such, the spectrum of a high altitude cloud will begin to deviate from the clear spectrum at a lower wavenumber than a lower altitude cloud. This is illustrated in Fig. 1a which also shows the spectra of three ash clouds of varying heights. Effectively, until the point where the clear and ash/cloudy spectra diverge, the instrument is recording clear radiances. This concept has been used to identify channels whose cloud free radiances can be assimilated into numerical weather prediction models, rather than filtering out these pixels entirely (e.g. McNally and Watts, 2003).

The changing sensitivity of each of the channels to the atmospheric profile is better demonstrated in Fig. 1b and c. This shows the derivative of atmospheric transmittance with log pressure (d$\tau$/dln$p$) and the peak of this value respectively. This is a measure of each channel's sensitivity to each level of the atmosphere and demonstrates that this shifts from the upper atmosphere at lower wavenumbers to the surface at higher wavenumbers.

As the channels are sensitive to different parts of the atmosphere it is possible to use this to estimate the height of the cloud (meteorological or in principle ash). To do this using the CO$_2$ slicing method, the ratio ($f$, Eq. 1) of the difference in cloudy and clear radiances ($L_{obs}$ and $L_{clr}$ respectively) for two channels ($\nu_1$ and $\nu_2$) within or close to the CO$_2$ absorption band is compared against a cloud pressure function ($C$, Eq. 2):

$$f(\nu_1, \nu_2) = \frac{L_{obs}\nu_1 - L_{clr}\nu_1}{L_{obs}\nu_2 - L_{clr}\nu_2} \tag{1}$$

$$C(\nu_1, \nu_2, p) = \frac{N\varepsilon_1 \int_{p_s}^{p_c} \tau(\nu_1, p) \frac{dB[\nu_1, T(p)]}{dp} dp}{N\varepsilon_2 \int_{p_s}^{p_c} \tau(\nu_2, p) \frac{dB[\nu_2, T(p)]}{dp} dp} \tag{2}$$



where $\tau$ is the atmospheric transmittance at channel $\nu$ emitted at pressure level $p$ arriving at the instrument; $B$ is the Planck radiance which is channel and temperature (and therefore pressure) dependant; $p_c$ and $p_s$ are the cloud and surface pressure respectively; and $N\varepsilon$ is the effective emissivity (sometimes referred to as the effective cloud amount), a product of the cloud fraction ($N$) and cloud emissivity ($\varepsilon$). Equation 1 is compared against Eq. 2 and where the two functions intersect is taken as

the cloud top pressure. A demonstration of this is shown in Fig. 2a. Following this the effective emissivity can be computed using a channel which falls within an atmospheric window ($w$; usually one close to $CO_2$ absorption band):

$$N\varepsilon = \frac{L_{\mathrm{obs}}(w) - L_{\mathrm{clr}}(w)}{B[w, T(p_c)] - L_{\mathrm{clr}}(w)} \tag{3}$$

In most applications of the $CO_2$ slicing technique, multiple channel pairs are used, resulting in different height solutions. In many studies, channel pairs are not considered if $L_{\mathrm{cld}}(\nu)$ - $L_{\mathrm{clr}}(\nu)$ for either the $CO_2$ ($\nu_1$) or reference ($\nu_2$) channels used falls

within the noise of the instrument at that channel (e.g. Menzel et al., 1992). The solution may also be rejected if the effective emissivity computed using Eq. 3 is not between 0 and 1.05 (e.g. Arriaga, 2007). If multiple solutions remain, then a number of different techniques can be employed to obtain a final value. This includes a top down approach where the solution of the most opaque channel is accepted if it is within an expected height range, and if not the next most opaque channel is considered. This is repeated until an appropriate height value is obtained (Menzel et al., 2008). Alternatively, the height and cloud fraction

which best satisfies the radiative transfer equation for all the channels used is accepted as the final cloud pressure/height (e.g. Menzel et al., 1983, 1992). If all of the channel pairs are considered inappropriate, for example, if $L_{\mathrm{cld}}(\nu)$ - $L_{\mathrm{clr}}(\nu)$ is within the noise of the instrument for all the channels used, then many methods assume that cloud is opaque and compare the brightness temperature measured by the instrument at 11 $\mu$m to an atmospheric temperature profile to obtain an alternative cloud height (e.g. Menzel et al., 1983, 1992; Zhang and Menzel, 2002; Menzel et al., 2008).

The issue of multiple solutions is further complicated for hyperspectral instruments as these can have hundreds of channels within the $CO_2$ absorption band. Some methods apply a weighting function based on each channel's sensitivity to the atmosphere (e.g Smith and Frey, 1990). However, to avoid a high computational cost, often there needs to be some prior consideration of the most appropriate channels. This has included exploring large datasets with known cloud top heights to select the most appropriate channels (e.g. Arriaga, 2007). Other approaches include the creation of synthetic channels by averaging

the radiances of channels sensitive to the same portion of the atmosphere (Someya et al., 2016) or $CO_2$ sorting which looks for the point where the clear and cloudy spectra deviate which is the first point where the instrument can see the cloud layer (Holz et al., 2006).

The $CO_2$ slicing method makes a number of assumptions: (1) That the cloud is infinitesimally thin; (2) the two channels used in Eq. 1 are sufficiently close that the difference in emissivity between them is negligible; (3) in cases where there are

multiple layers of cloud, the lower level clouds are ignored. The second is particularly important to consider when the channel pairs are selected. Multiple cloud layers have previously been identified as a source of error in the $CO_2$ slicing retrieval with the extent of this being affected by the channels used and the height of the underlying layers (Menzel et al., 1992). For example, an opaque cloud close to the surface is unlikely to affect the height retrieval of a cirrus cloud when using channels which are





not sensitive to radiation from the lower troposphere. In contrast, a opaque cloud in the middle of the troposphere might lead to the underestimation of the cloud top height of a higher cirrus layer (Menzel et al., 1992). The effect of surface emissivity is expected to be minimal as channels within the $CO_2$ absorption band have weighting functions which peak above the surface, as shown in Fig. 1d.

An additional consideration has to be made when applying the $CO_2$ slicing method to volcanic ash. The height that a volcanic ash cloud reaches is largely dependant on the force of the eruption and the atmospheric conditions (Sparks et al., 1997) and so this can vary widely. Large explosive eruptions can generate columns which enter the stratosphere, which can then potentially affect climate (Robock, 2000). The cloud pressure function generated using Eq. 2 is temperature dependant. Within the troposphere, the temperature decreases with height; however, in the stratosphere the temperature beings to climb

again. This leads to a reversal in the cloud pressure function, which in some cases can result in multiple solutions: one in the troposphere and one in the stratosphere. Consequently, some prior information is required to determine whether the plume is within the troposphere and therefore if the $CO_2$ slicing technique is appropriate. This might include observations made on the ground or by pilots. The $CO_2$ slicing technique has previously only been used to determine the height of meteorological clouds in the troposphere and so in this study only the tropospheric solution is accepted.

## 3   The Infrared Atmospheric Sounding Interferometer

The Infrared Atmospheric Sounding Interferometer (IASI) is an instrument on-board three meteorological satellites, MetOp A, B and C, launched in 2006, 2012 and 2018 respectively. Each instrument orbits the Earth twice a day. The instrument scans have a swath width of 2200 km and consist of groups of four circular pixels which have a diameter of 12 km at nadir (Clerbaux et al., 2009). The instruments measure across the infrared between 645 to 2760 $cm^{-1}$ (3.62 to 15.5 $\mu$m) with a high spectral

resolution of 0.5 $cm^{-1}$ (Blumstein et al., 2004).

    The instrument has previously been used to analyse volcanic plumes of $SO_2$ (e.g. Clarisse et al., 2008; Walker et al., 2012; Carboni et al., 2012; Clarisse et al., 2012, 2014; Carboni et al., 2016; Taylor et al., 2018) and ash (e.g. Clarisse et al., 2010; Maes et al., 2016; Ventress et al., 2016) from a number of different eruptions. Previous methods for determining the height of the plume with spectra measured by IASI use the optimal estimation method (Maes et al., 2016; Ventress et al., 2016). The $CO_2$

slicing method has previously been applied to IASI spectra to obtain the cloud top height of meteorological cloud (Arriaga, 2007). The values obtained for the cloud pressure and emissivity are often assimilated in numerical weather prediction models (Guidard et al., 2011; Lavanant et al., 2011). The different adaptations of the $CO_2$ slicing technique for IASI use different numbers and combinations of channels and can therefore give different results (Lavanant et al., 2011). In this study, channels are selected based on the technique's performance when applied to simulated ash spectra.



## 4 Application to simulated data

### 4.1 Channel selection

IASI has over 300 channels which fall within the $CO_2$ absorption band, and so, to ensure computational efficiency an appropriate subset of these channels must be selected. To do this the $CO_2$ slicing technique was first applied to 384 simulated ash
spectra. These spectra include six different atmospheres: high latitude, mid-latitude day and night, tropical daytime and polar summer and winter. The simulated spectra also represent a range of different ash properties: ash optical depths ranging between 5 and 15 (referenced at 550 nm), ash effective radius between 5 and 10 $\mu$m and cloud heights from 200 to 900 mb.

The $CO_2$ slicing method was first applied using every channel combination between 660 and 800 cm$^{-1}$, where the reference channel ($\nu_2$) is greater than the $CO_2$ channel ($\nu_1$). In this way, the reference channel is generally more sensitive to a lower
part of the atmosphere than the $CO_2$ channel. As with existing studies only tropospheric solutions were accepted and in cases where the curve of the cloud pressure function resulted in multiple solutions, the solution with the greater weight (in this case the weighting function is defined as $w = d\tau[\nu_1, p]/dlnp$) was accepted. The output from each channel pair was only accepted if it met three quality control criteria: (1) $L_{cld}\nu_1$ - $L_{clr}\nu_1$ must be greater than the noise of the instrument at channel $\nu_1$ ($CO_2$ channel); (2) Similarly, $L_{cld}\nu_2$ - $L_{clr}\nu_2$ must be greater than the noise of the instrument at $\nu_2$ (reference channel); (3) The
solution to Eq. 3 must fall between 0 and 1.05 (following Arriaga 2007).

The results are shown in Fig. 3. The top two lines show the maximum pressure difference between the true (simulated) and $CO_2$ slicing retrieved values divided into each pressure level. In total there are 48 spectra for each pressure level with these incorporating the different atmospheric profiles and ash properties. The lower two lines of Fig. 3 show the percentage of accepted retrievals. This refers to where there was an intersection between the two functions shown in Eq. 1 and 2, and where
the value retrieved meets all three quality control conditions. This is also grouped into the eight pressure levels. The equivalent plots for the six individual atmospheres can be seen in Fig. A1-A6 in the appendix. Potentially, the method used in this study to select the most appropriate channels, could be performed for the different atmospheres to select channels which might be more suited to specific climatologies.

Figure 3 demonstrates that the best performing channels vary depending on the height of the plume. As expected, this shifts
from lower wavenumbers at lower pressures to higher wavenumbers closer to the surface. Notably, at 200 mb there are far less channels which pass the quality control conditions, and where a retrieval is possible, there is a large difference between the true and retrieved pressure. It is also possible to identify an increased error closer to the surface. Previous studies have acknowledged that the $CO_2$ slicing tool is less successful at pressures greater than 700 mb (Menzel et al., 2008) because approaching the surface there are less channels with a distinction between the clear and cloudy spectra, often leading to $L_{cld}(v)$
- $L_{clr}(v)$ to be within the range of the instrument's noise and therefore the channels being excluded. Another observation that can be made from Fig. 3 is that channels below 700 cm$^{-1}$ often have a low percentage of accepted retrievals. These channels are shown in Fig. 1b and c to be sensitive to the heights above the tropopause. This may also be the reason for few accepted retrievals at 720 cm$^{-1}$. Additionally, for channels greater than 750 cm$^{-1}$, which are no longer in the $CO_2$ absorption band, the difference between the true and retrieved pressure is usually greater than 100 mb.





Figure 4 shows a similar plot between 700 and 750 cm$^{-1}$. In this case, the spectra were also grouped into three categories: high cloud (300-400 mb), mid level cloud (500-600 mb) and low level cloud (700-800 mb). Note that the simulated spectra at 200 and 900 mb have been excluded. Also, the maximum pressure difference is only shown where it is less than 75 mb and where the percentage of successful retrievals is greater than 50%. This plot has been used to manually select the most

appropriate set of channels. The best selection of channel pairs will be representative of the entire atmosphere (channels should be selected which peak at different heights (Fig. 1c), while minimising the difference between the simulated and retrieved pressures, and maximising the acceptance rate, Fig. 4. Another consideration is the assumption that the change in emissivity between the channel pairs is negligible. The emissivity ratio for a sample of ash from the Eyjafjallajökull eruption (the main eruption considered in this study) for all channel combinations in the 680 and 800 cm$^{-1}$ range is shown in Fig. 5. For this

assumption to hold true, the emissivity ratio should be as close to 1 as possible. This is usually the case for channels which are close together. Given these criteria appropriate channel ranges have been selected. These channel ranges and the reference channels are shown in table 3. The weighting functions for the selected channels are shown in Fig. 1d.

## 4.2    Simulation results

Following the selection of channels, the final pressure values ($P$) were computed by taking a weighted average of the results:

$$P = \frac{\sum p_c(\nu) w^2(\nu)}{\sum w^2(\nu)} \qquad (4)$$

where $p_c$ is the pressure retrieved for channels $\nu$ and $w$ refers to the weighting function ($d\tau[\nu, p]/dlnp$). On this occasion, the retrieval was applied to 1344 simulated ash spectra including those with lower ash optical depths (ranging from 0.5 to 15) and smaller effective radius (ranging from 1 to 10 $\mu$m). This includes spectra representative of thinner ash clouds which were not considered during the channel selection.

The results are displayed in Fig. 6a-f which plots the true (simulated) pressures against the final weighted pressures obtained with the CO$_2$ slicing technique. The different atmospheres are displayed separately and the percentage of accepted retrievals are indicated below each plot. Table 4 reports the root mean square error (RMSE) for each atmosphere. Overall, the CO$_2$ slicing method returned values for 72% of the simulated spectra, with an RMSE of 777 m. These results suggest that the technique does have merit for obtaining the height of ash clouds.

Figures 6g-i give some indication of where and why the retrieval was unsuccessful. Figure 6g-h show there are slightly more failed cases for ash spectra with the lowest optical depth (0.5) and effective radius (1 $\mu$m). These low values are representative of thinner ash clouds whose spectra are more similar to clear atmospheric spectra. Subsequently, these cases are likely to fail the signal/noise quality control tests (Menzel et al., 1992, 2008). For example, an ash cloud at 500 mb only has 7 channels which pass the $L_{cld}\nu_1$ - $L_{clr}\nu_1$ quality control condition when the ash optical depth is 0.1. However, the number of channels

passing this criterion increases to 38 at an ash optical depth of 2.3. The majority of failed cases are shown to be at the pressure extremes, Fig. 6i. Close to the surface this can again be attributed to less distinction between the clear and ashy spectra (Menzel et al., 2008). For example, for the RTTOV default atmosphere, an ash plume at 900 mb fails the signal/noise condition for all



the channels used regardless of the optical depth and effective radius of the simulation. The lowest simulation pressure (200 mb) is close to or above the tropopause for all six atmospheres and for this example the $CO_2$ slicing method was allowed to retrieve up to the height of the reversal of the temperature profile (which is slightly above the tropopause). At these heights, the temperature gradient ($dT/dp$) is relatively stable, causing a similar effect in the cloud pressure function (best illustrated

in Fig. 2) and subsequently a greater number of unsuccessful retrievals: the $CO_2$ slicing technique has previously been shown to perform poorly in isothermal regions of the atmosphere (Richards et al., 2006). This may also be the reason for the poor performance of the $CO_2$ slicing technique when applied to the polar summer atmosphere for which the technique only retrieved values for 29% of cases.

The RMSE and the percentage of accepted retrievals for the $CO_2$ slicing technique, without the quality control criteria

applied, are shown in table 4. Figure A7 shows the equivalent plot to Fig. 6 without the quality control. The addition of the quality control compromises the number of successful retrievals for an overall reduction in the RMSE. Overall, the reduction is around 200 m but in individual cases by up to 1.4 km (e.g. tropical atmosphere). Figure A7 indicates that the addition of the quality control is particularly advantageous for lower level ash layers which without the quality control are often overestimated. Overall, the results show that this adaptation of the $CO_2$ slicing technique has promise for obtaining the height of volcanic ash

clouds within the troposphere, although its use is limited in cases of low level or thin clouds or where there is a steep temperature gradient.

## 5    Application to scenes containing volcanic ash

The $CO_2$ slicing method has been applied to scenes containing ash from the Eyjafjallajökull (63.63°N, 19.63°W, 1651 m) and Grimsvötn (64.42°N, 17.33°W, 1725 m) eruptions in 2010 and 2011 respectively. The plumes from both eruptions were closely

monitored using a variety of instrumentation which included ground based remote sensing, airborne measurements and the use of satellite products (e.g. Gudmundsson et al., 2010; Weber et al., 2012). The ash and gas clouds from these eruptions have since been extensively studied (e.g. Kerminen et al., 2011; Tesche et al., 2012; Flemming and Inness, 2013; Cooke et al., 2014; Ventress et al., 2016). They are commonly used to demonstrate the utility of new remote sensing developments (e.g. Mackie and Watson, 2014; Taylor et al., 2015; Ventress et al., 2016; Western et al., 2017), and similarly are often used in modelling

research (Matthias et al., 2012; Webster et al., 2012; Moxnes et al., 2014; Wilkins et al., 2016). This makes them the ideal first candidates for the $CO_2$ slicing technique. Another reason for choosing these eruptions is that in both cases, the ash clouds were confined to the troposphere making them an appropriate target for the $CO_2$ slicing technique.

In this application of the retrieval, it has only been applied to pixels which are flagged as containing volcanic ash by a linear ash retrieval developed for IASI (Ventress et al. 2016: following the method developed for $SO_2$ by Walker et al. 2012). Values

for $L_{clr}$ were obtained using the radiative transfer model RTTOV. The temperature and humidity profiles needed to calculate the planck radiance and $\tau$ were acquired from the European Centre for Medium-Range Weather Forecasts (ECMWF). The closest ECMWF profile to each individual IASI pixel was used. RTTOV was used to compute the transmittance values. Another point to note is that, in section 4, the maximum height that could be retrieved was defined as the height at which the temperature



profile inverts and has a positive gradient. Figure 6 demonstrated that the $CO_2$ slicing method performs poorly where the temperature profile steepens significantly. For the application to real satellite data, the maximum height which can be retrieved is the height of the tropopause as defined by the World Meteorological Organisation: the point at which the lapse rate is less than 2°C, and remains lower than this for at least 2 km.

## 5.1 Methods used for comparison

### 5.1.1 Optimal Estimation Scheme

The $CO_2$ slicing plume altitude results have been compared against the plume altitude obtained using the optimal estimation (OE) retrieval scheme developed by Ventress et al. (2016). The retrieval scheme combines a clear-sky forward model with a (geometrically) infinitely thin ash layer to simulate atmospheric spectra, using ECMWF data as input atmospheric parameters. The simulated spectra are compared to the satellite measurements and, using the cost function (a measure of retrieval fit), the spectrum that most closely matches the spectrum obtained with IASI is used to determine the ash plume properties. This method retrieves the effective radius and ash optical depth, which can be used to calculate the mass of ash within the plume. For more information on this technique, refer to Ventress et al. (2016).

### 5.1.2 CALIOP

While a comparison against another IASI retrieval is useful, such comparisons have limitations. All retrieval techniques make assumptions and have different limitations and so it is not expected that the results would be the same, or even similar, in all cases. An additional comparison is made with the Cloud-Aerosol LiDAR with Orthogonal Polarization (CALIOP) instrument, on-board the the Cloud-Aerosol LiDAR and Infrared Pathfinder Satellite Observations (CALIPSO) satellite. This active sensor was launched in 2006 and forms part of NASA's afternoon constellation (A-Train) of satellites. The instrument has a 30 m vertical resolution and 335 m spatial resolution, and orbits roughly every 16 days (Winker et al., 2009; Hunt et al., 2009). The backscatter profile obtained with LiDAR instruments can be used to obtain the vertical structure of the atmosphere, providing information on the height and thickness of different scattering layers, including both ash and cloud. CALIOP and other LiDAR instruments are commonly used as a tool for the validation of cloud heights, including the $CO_2$ slicing technique (e.g. Smith and Platt, 1978; Frey et al., 1999; Holz et al., 2006, 2008) and a number of ash retrievals (e.g. Stohl et al., 2011; Ventress et al., 2016).

To conduct a comparison between the heights obtained using the $CO_2$ slicing and OE techniques with CALIOP the data from the two instruments was first collocated. CALIOP overpasses which intersected with the ash plumes were identified using false colour images from the Spinning Enhanced Visible and Infrared Imager (SEVIRI) (Thomas and Siddans, 2015). The backscatter profiles were then averaged vertically to a 250 m resolution. The CALIOP data was smoothed to IASI's spatial resolution of 12 km and collocation was identified where measurements made by the two instruments fell within 50 km and 2 hours of each other. If multiple CALIOP pixels were matched to an IASI pixel then the CALIOP pixel which was closest in distance was selected for comparison. A cloud top height is obtained from the backscatter profiles allowing a comparison with





the $CO_2$ slicing and OE methods. This was done by (1) calculating the mean backscatter above 15 km and subtracting this from the total backscatter; (2) for each pixel a cumulative backscatter is calculated; (3) the cloud altitude is where the atmospheric extinction exceeds a specified threshold. This threshold has been manually set for each scene, chosen to obtain the best match to the cloud top height shown in the CALIOP backscatter images.

## 5.2   Comparison of results

The $CO_2$ slicing technique was applied to IASI ash flagged pixels from 13 and 4 days from the Eyjafjallajökull and Grimsvötn eruptions respectively. Maps of these results, with the orbits divided into morning and afternoon are shown in Fig. 7. For each map there is a histogram showing the distribution of the retrieved heights. Encouragingly, initial examination of the maps shows that the retrieved values are spatially consistent with only a few outliers. These outliers are usually individual pixels whose altitudes are higher than those surrounding them. Below each map are numbers indicating the total number of pixels in each plot and the number of pixels for which the $CO_2$ slicing technique was unable to obtain a height, either because there is no intersection between the two functions shown in Eq. 1 and 2 or because of the failure of one or more of the quality control measures outlined in section 4. Overall, the $CO_2$ slicing technique was able to obtain a height value for 88% of pixels from the two eruptions.

The $CO_2$ slicing results have been compared against those obtained with an optimal estimation (OE) scheme. Distributions of the heights obtained for all pixels from the two eruptions are shown in Fig. 8a and b. In both cases, the peak of the distribution for the $CO_2$ slicing heights is higher than for the OE scheme. Figure 9 shows how the average height obtained with the two retrievals has changed over the 13 days studied from the Eyjafjallajökull eruption. This plot shows that on the $5^{th}$ May the $CO_2$ slicing method retrieved an average altitude of roughly 7 km and that this then fell throughout the remainder of the study period. This corresponds to observations made about the volcano's activity. Activity at the volcano became more explosive on the $5^{th}$ May 2010 with increased emission of ash and $SO_2$, with plumes rising to greater than 8 km. This was followed by a fall in the plume height to 6-7 km: interspersed with higher plumes during more explosive activity (Petersen, 2010). The average $CO_2$ slicing heights shown in Fig. 9 are lower because these are values for the entire plumes including further away from the source. However, it does capture the changing elevation of the plume throughout the eruption. By contrast, the OE average heights are less variable: between 3 and 4.25 km throughout the period studied. Some example maps of the OE results are shown in Fig. 10 to 13. The different assumptions and limitations of the two techniques mean that it is not expected that the two retrievals will return the same or even similar values. The persistently lower average height for the OE technique suggests that it is strongly influenced by the height a priori (which was around 3.5 km). In future applications of the OE scheme, the $CO_2$ slicing results could be used as the a prioi. Other differences in the results may arise from the nature of the two techniques. The OE scheme returns values for the ash optical depth, effective radius and height by fitting simulated spectra to those obtained with IASI. Ventress et al. (2016) identified that in some cases the retrieval assumed a lower altitude and a higher ash optical depth in order to fit the spectra. Additional differences may arise from the channels used. As explained in section 4, the channels used for the $CO_2$ slicing have been specifically chosen for their ability to obtain the ash cloud height of simulated data. In contrast, Ventress et al. (2016) suggested that the OE height retrieval could be further refined by altering the channels used. One suggestion was





to select channels which minimise the effect of the underlying cloud layers following observations that the OE method can underestimate the cloud top height in cases of multiple cloud layers (Ventress et al., 2016).

A comparison has been made against backscatter profiles and cloud altitudes obtained with CALIOP, to assess how successfully the two retrievals perform. These backscatter profiles are shown in Fig. 10-13d. The heights obtained from the OE and $CO_2$ slicing methods for pixels which fall within 2 hours and 50 km are overplotted, along with the heights obtained with CALIOP and the tropopause height. In these plots it is possible to observe that both methods are capable of capturing the height of the ash layer, but there are clear cases where one technique outperforms the other. In Fig. 10 which shows the backscatter plot for the $6^{th}$ May 2010, the $CO_2$ slicing method places the ash cloud between 5 and 7 km between 57.5 and 60.5°N. This is shown to be higher than the CALIOP heights (4-5 km) to which the OE results are a closer match. In the same image, between 63 and 64°N the $CO_2$ slicing results are again higher than the OE results but this time are closer to, but lower than, the heights obtained from CALIOP. The lower heights of both the $CO_2$ slicing and OE scheme relative to CALIOP might be related to the thick underlying cloud layer. Figure 11d shows another example from the $9^{th}$ May 2010. Here between 51 and 53°N the heights obtained with both methods match those obtained with CALIOP. However, further north between 56 and 60°N, the $CO_2$ slicing results agree more closely compared to those from the OE scheme. At 66°N the $CO_2$ technique obtains a value close to the cloud top height, whereas the OE scheme obtains a value which is more representative of a lower layer of cloud. Figures 12 and 13 shows examples from the Grimsvötn eruption and in both cases both height retrievals are shown to resemble the shape of the ash cloud layer shown by CALIOP. There are cases where both retrievals underestimate the cloud top height which may be due to multiple layers of cloud.

Pearson's correlation values and the root square mean error (RMSE) were computed to compare the two retrieval methods against the heights obtained with CALIOP. These are shown in table 5 and scatter plots comparing the retrieved values are shown in Fig. 8c and d. The Pearson's correlation values are greater for the $CO_2$ slicing than for the OE scheme, while the RMSE values are lower: 2.2 and 2.1 km for the the Eyjafjallajökull and Grimsvötn eruptions respectively for the $CO_2$ slicing technique, compared to 3.2 and 2.4 km obtained for the OE method. This implies an improved height retrieval from the $CO_2$ slicing method.

Although comparisons against LiDAR backscatter profiles are a common way of validating retrievals of ash and meteorological cloud height, these comparisons can be limited. CALIOP and IASI measure different things. The first measures backscattering while the latter measures thermal emission. Measurements are made with significantly different spatial resolutions (335 m compared to 12 km for CALIOP and IASI respectively) and in different locations (a maximum difference of 50 km). Clouds can also vary significantly in very short spaces of time. Although only pixels with a difference of 2 hours have been considered in this comparison, this is still sufficient time for changes in the cloud's position both vertically and horizontally. These may account for some of the differences seen between the CALIOP profiles and the results obtained with the $CO_2$ slicing and the OE scheme. The cloud heights obtained from the CALIOP profile are not always a perfect representation of the cloud top height which may also contribute to the differences observed. Although these limitations exist, comparisons against LiDAR instruments are still one of the best methods for validating cloud heights, and in this case demonstrate that the $CO_2$ slicing technique has potential as a tool for obtaining the cloud top height of volcanic ash.



## 6 Conclusions

The $CO_2$ slicing technique is an established method, used for decades, for retrieving the cloud top height of meteorological cloud. Although it has previously been acknowledged that it can be applied to volcanic ash, it is not commonly used for this purpose, and it has only been applied to MODIS. In this study, the technique was adapted for IASI using simulated ash data to select the most appropriate channels and then demonstrate the technique's capability. When applied to the simulated data, the technique was shown to perform well in five out of six atmospheres. However, an increased failure rate, was seen above and close to the tropopause and close to the surface. This was also true of ash with lower optical depths and effective radius. Similar observations have been made by previous $CO_2$ slicing studies. In this application three quality control criteria have been applied which successfully remove the majority of cases where there are large differences between the true and retrieved pressures. When applied to ash scenes from the Eyjafjallajökull and Grimsvötn eruptions, the $CO_2$ slicing results compared well against the CALIOP backscatter profiles. It was also demonstrated that the $CO_2$ slicing method obtained heights which more closely matched CALIOP than the optimal estimimation scheme used for comparison.

This is the first application of the $CO_2$ slicing technique to obtain the height of volcanic ash from IASI spectra, and the results are very encouraging. One advantage of this algorithm is that it can be run fairly quickly and so it could be applied to get a first approximation of the height, which could then be used to help assist hazard mitigation. It can also then be used as an input parameter into models of ash cloud propagation or as an a priori in other retrieval schemes. There is also potential for the further development of this technique in the future. Previous applications to cloud have created synthetic channels (multiple channels averaged together) which could be used to further improve the algorithm and its sensitivity to lower level clouds (Someya et al., 2016). It would also be possible to explore other options for selecting channels or obtaining the final cloud height. The channel selection in this study was based on simulated data in six different atmospheres, another avenue to explore would be the selection of atmospheric specific channel pairs. Further work would also help appreciate the strengths and limitations of this technique, and therefore where its use is most appropriate.

*Data availability.* The data used in this paper can be made available by contacting the author (isabelle.taylor@earth.ox.ac.uk)

## Appendix A

Some additional figures are included within this appendix. Figures A1 to A6 show the maximum difference between the true (simulated) and retrieved pressures for the six investigated atmospheres for all the channel combinations between 660 and 800 $cm^{-1}$. The plots are divided into the different pressure levels. The figure also includes the percentage of successful retrievals (where there is an intersections between the two functions shown in Eq. 1 and 2 and all quality control conditions are met). This is out of a total of 8 simulations (for each pressure level) with ash optical depths ranging between 5 and 15, effective particle radius ranging between 5 and 10 $\mu$m. These could be used to select channels which are appropriate for specific climatologies. Figure A7 shows the final simulation result for each atmosphere without the quality control applied.





*Competing interests.* There are no competing interests at present

*Acknowledgements.* IAT, EC, TAM and RGG were supported by the NERC Centre for Observation and Modelling of Earthquakes, Volcanoes, and Tectonics (COMET). IAT's research is funded by the NERC Environmental Research DTP (NE/L002612/1). IAT was also supported by a Met Office Academic Partnership.

5    Five of the atmospheric profiles used for the channel selection are reference spectra for the Michelson Interferometer for Passive Atmospheric Sounding (MIPAS; Remedios et al. 2007). The IASI spectra used in this study are available from the Centre for Environmental Data Analysis (EUMETSAT, 2009). Atmospheric profiles needed to run the $CO_2$ slicing technique were obtained from the European Centre for Medium-Range Weather Forecasts (ECMWF).



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





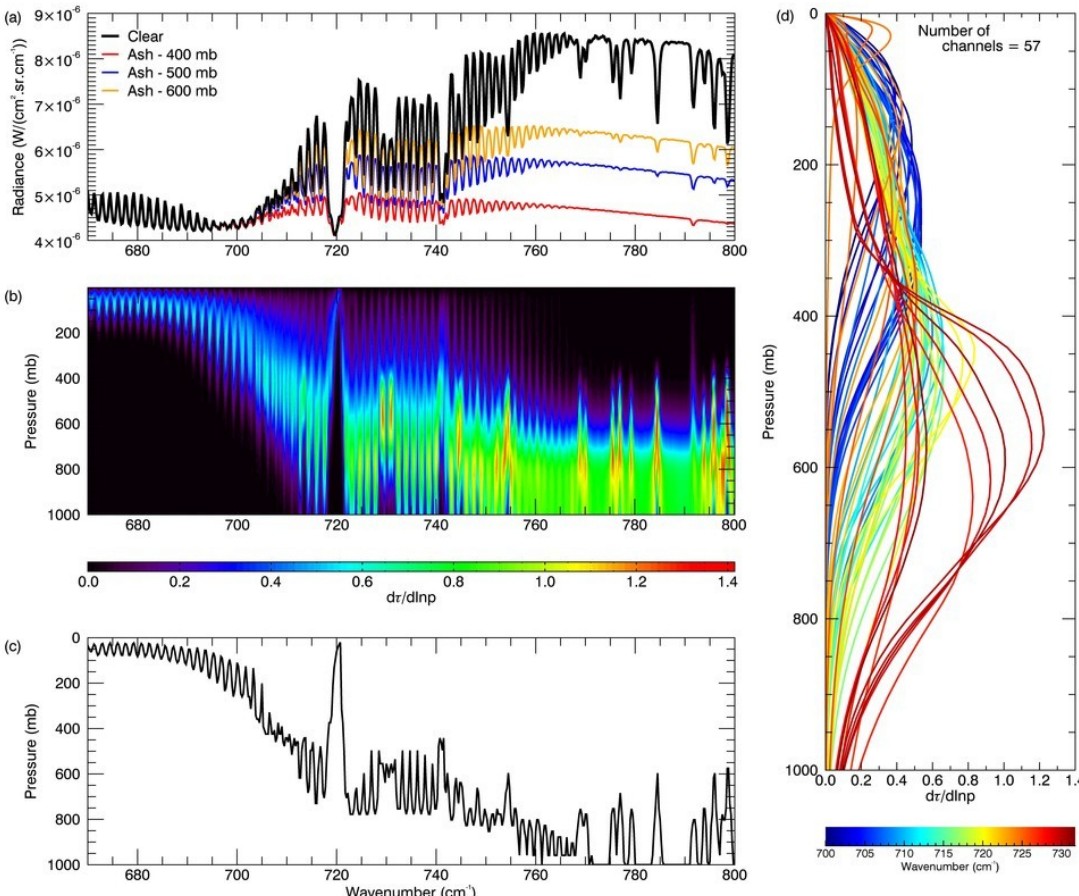

**Figure 1.** (a) Simulated spectra for a clear atmosphere (i.e. one without cloud or ash) and three ash clouds at different pressure levels: 400, 500 and 600 mb. (b) The change in atmospheric transmittance with log pressure ($d\tau/dlnp$). This is indicative of which part of the atmosphere each channel is sensitive to. This sensitivity is shown to shift from higher up in the atmosphere to the lower parts of the atmosphere as wavenumber increases. (c) The peak sensitivity for each channel. (d) The weighting function ($d\tau/dlnp$) for the 57 channels used in this $CO_2$ slicing study.





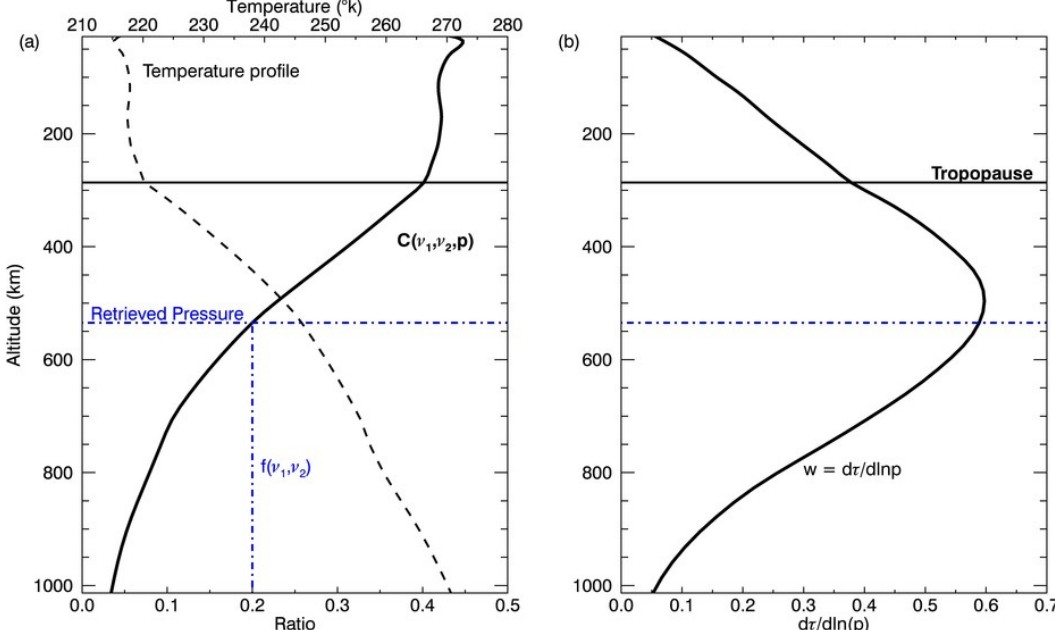

**Figure 2.** (a) An example of the cloud pressure function calculated using Eq. 2. This is strongly linked to the atmospheric temperature profile (dashed black line). The value obtained with Eq. 1 is compared against the cloud pressure function and where these intersect is taken as the cloud pressure solution for that channel. (b) The corresponding weighting functions ($d\tau/dlnp$) which illustrates the changing sensitivity to the atmosphere. This is used to obtain a weighted average from multiple channel solutions.





**Figure 3.** $CO_2$ slicing results for simulated ash spectra. The technique has been applied for each channel pair between 660 and 800 cm$^{-1}$. A total of 384 spectra were used which includes six different atmospheres. It also includes ash optical depths between 5 and 15, effective radius ranging between 5 and 10 $\mu$m and pressures between 200 and 900 mb. The first two lines of the plot show the maximum difference between the known (simulated) pressure and the pressure retrieved with the $CO_2$ slicing algorithm. This is divided into each pressure level. The last two lines show the percentage of successful retrievals. This is again divided into the 8 different pressure levels.





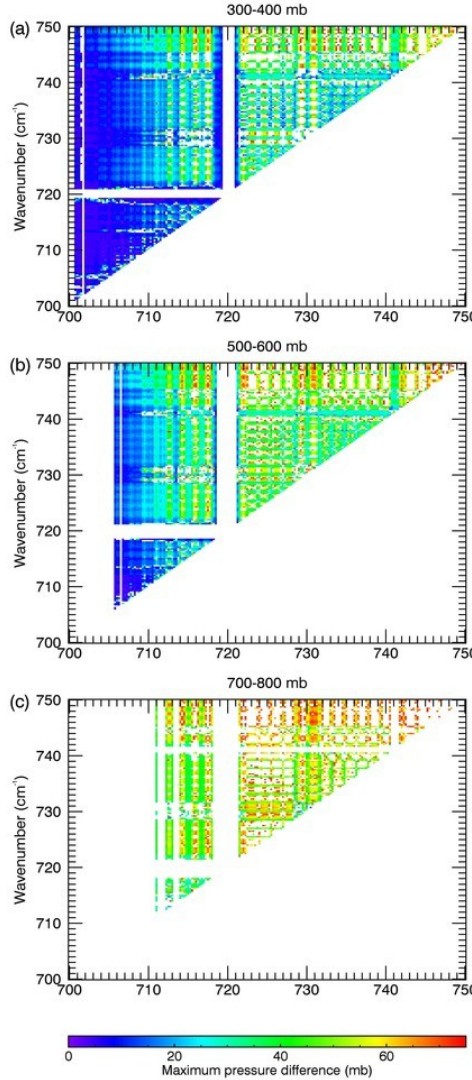

**Figure 4.** $CO_2$ slicing results for RTTOV simulated ash spectra. The plots show the maximum difference between the true (simulated) pressure and the pressure obtained with the $CO_2$ slicing algorithm. The results are split into three pressure levels: (a) high cloud (300-400 mb), (b) mid level cloud (500-600 mb) and (c) low level cloud (700-800 mb). Note that in this plot, results for 200 and 900 mb are excluded. Results are only included where the maximum difference is less than 75 mb and the percentage of successful retrievals is greater than 50%. This is used to inform the choice of channels for the final $CO_2$ slicing algorithm.





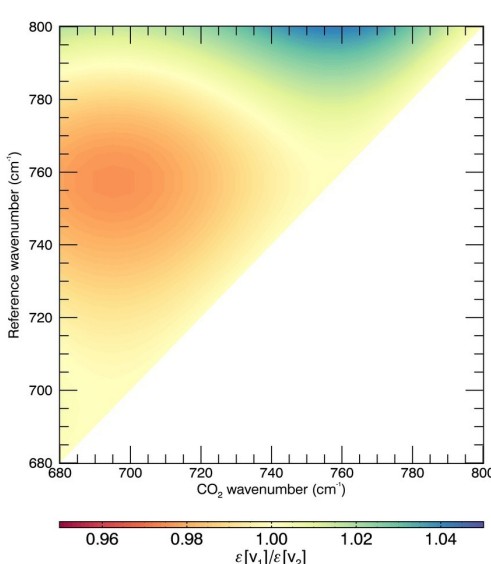

**Figure 5.** Emissivity ratio for channels between 680 and 800 cm$^{-1}$. The ash sample was from the Eyjafjallajökull eruption in 2010. The assumption that the emissivity does not vary significantly for the pair of channels used for the CO$_2$ slicing is important. For this to hold true, ideally the emissivity ratio should be close to 1.



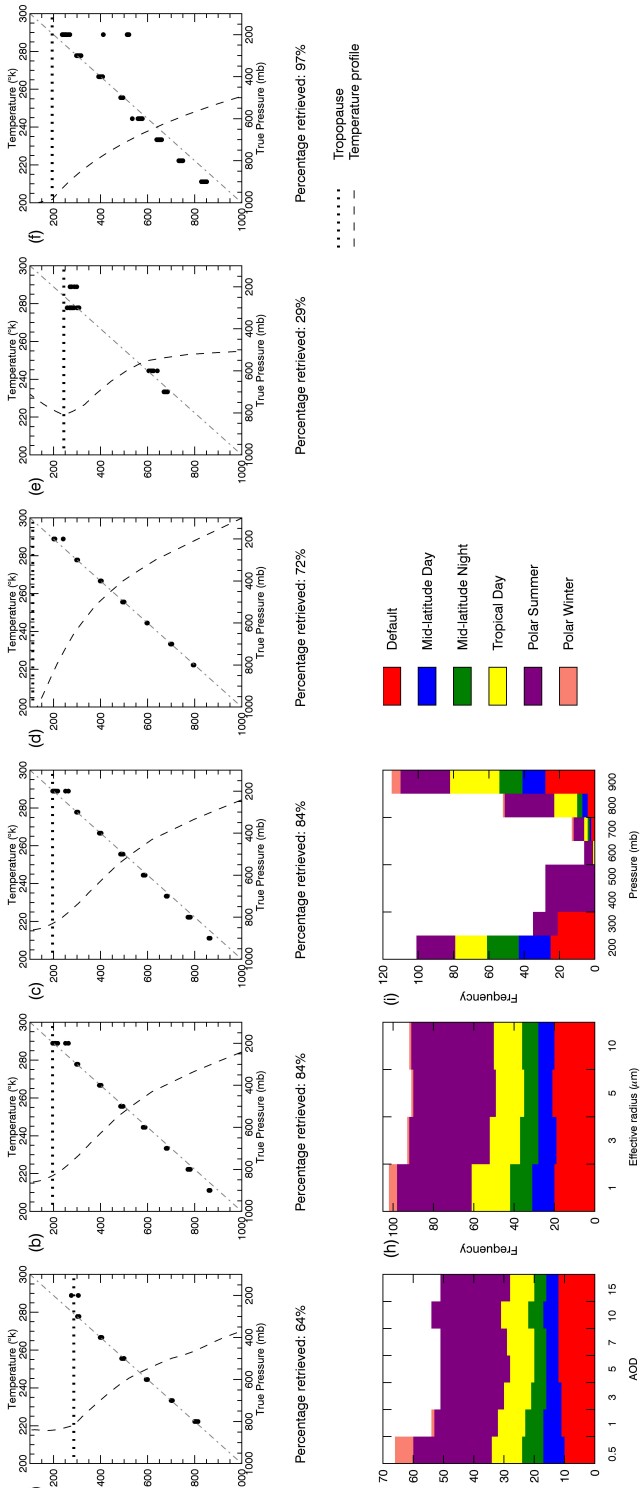

**Figure 6.** Final $CO_2$ slicing pressure results for RTTOV simulated ash spectra (a total of 224 spectra per atmosphere). The plots show the true (simulated) pressure plotted against the $CO_2$ slicing retrieved value for the six different atmospheres. (a) RTTOV default atmosphere (high latitude), (b) Mid-latitude day, (c) Mid-latitude night, (d) Tropical day, (e) Polar summer (f) Polar winter. In this case, the simulated spectra include the following ash properties: ash optical depth ranging between 0.5 and 15, ash effective radius ranging between 1 and 10 $\mu$m and pressure values between 200 and 900 mb. Below each plot is a value indicating the percentage of successful retrievals (where a height value can be obtained and all quality control conditions have been met). (g) the frequency of ash optical depths for which the $CO_2$ slicing technique was unable to return a height value. (h) Same as (g) for the effective radius. (i) Same as (g) for the ash cloud pressure. Related statistics can be seen in table 4. The equivalent plot, where the values which have not met the quality control conditions has been included in the appendix, figure A7.



**Figure 7.** Maps of the $CO_2$ slicing output (with quality control applied) for the Eyjafjallajökull and Grimsvötn eruptions. Each plot consists of multiple orbits, divided into morning and afternoon. On each plot is a histogram showing the distribution of heights for each scene. Beneath each plot are numbers showing the total number of pixels in each image and the number of pixels for which the $CO_2$ slicing method was unable to return a value.





**Figure 7.** *continued*



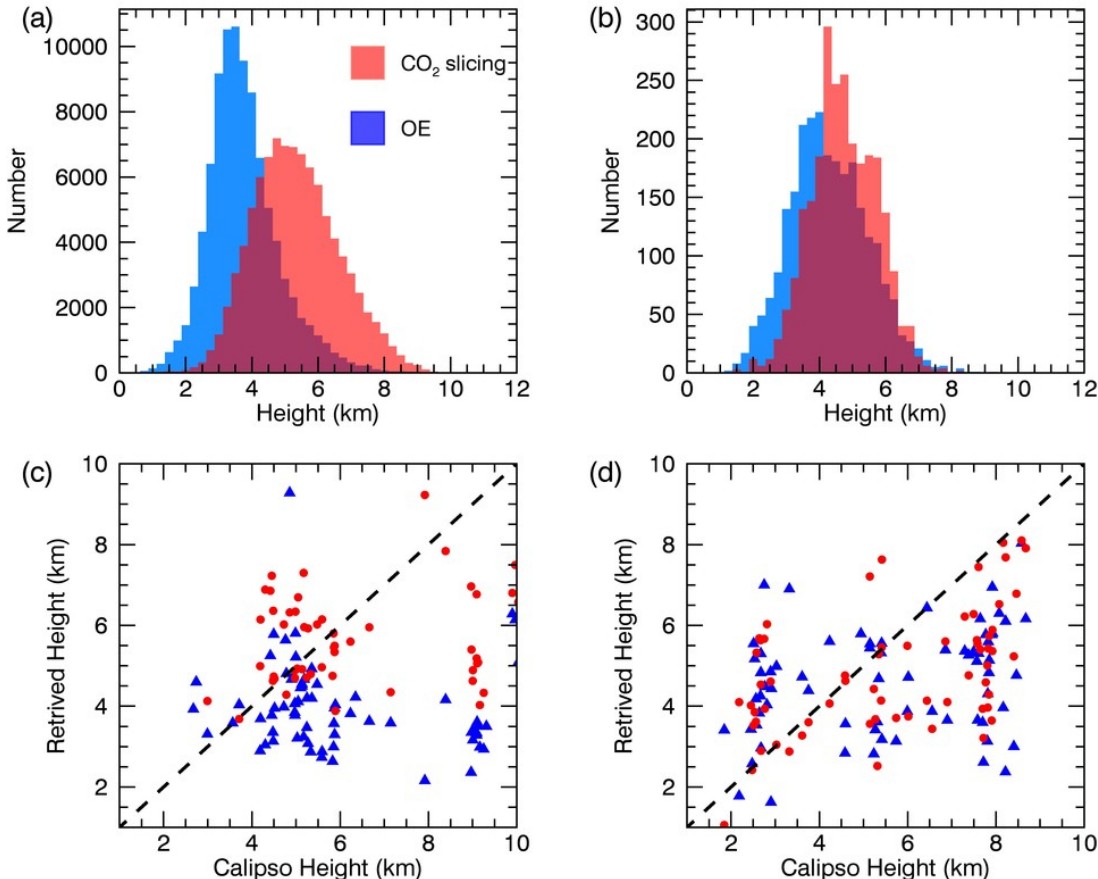

**Figure 8.** (a) Distribution of the $CO_2$ slicing and optimal estimation retrieved ash heights for the Eyjafjallajökull eruption. (b) Distribution of the retrieved ash heights for the Grimsvötn eruption. (c) Comparison of the CALIOP heights with those obtained with the $CO_2$ slicing and optimal estimation techniques for the Eyjafjallajökull eruption. (d) Same as (c) for the Grimsvötn eruption. Related statistics can be seen in table 5.





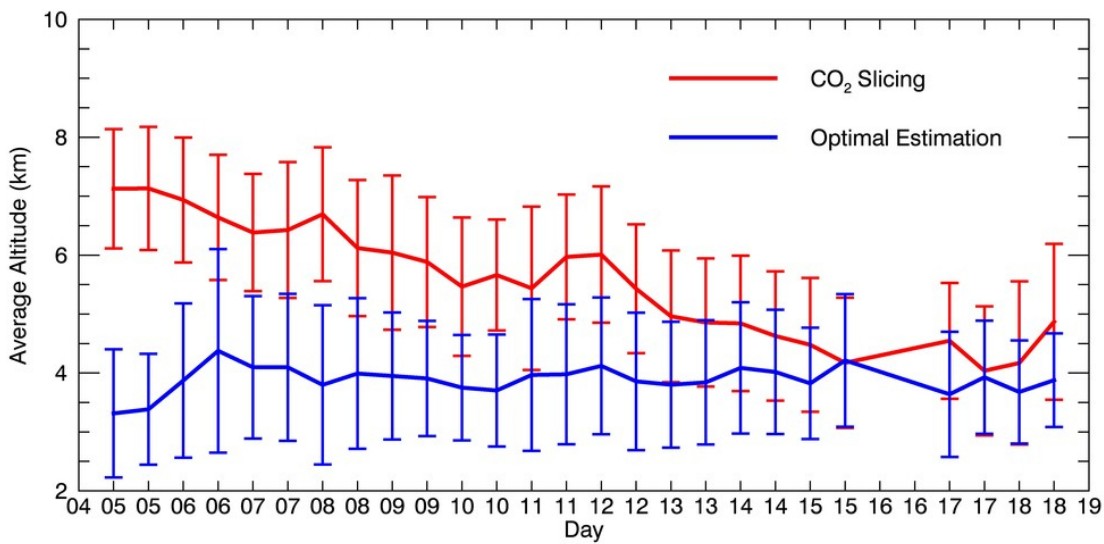

**Figure 9.** Time series showing how the average retrieved height for the $CO_2$ slicing and optimal estimation techniques varies during the Eyjafjallajökull eruption.





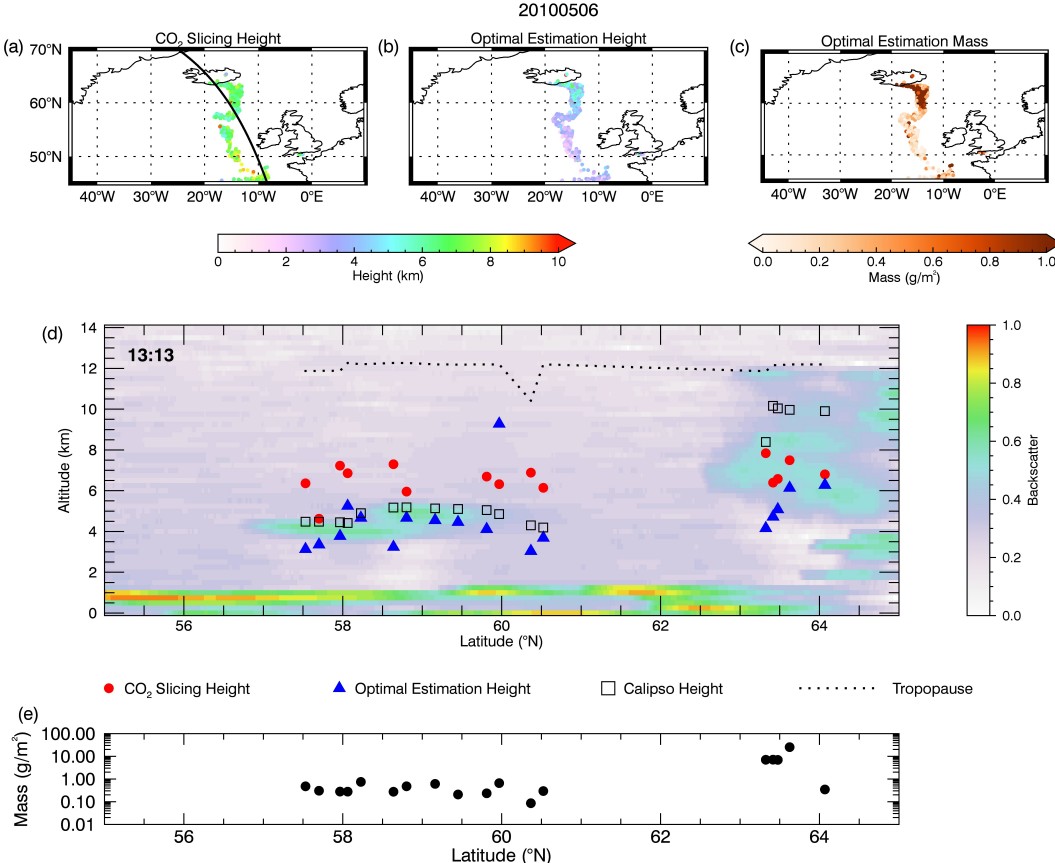

**Figure 10.** (a) $CO_2$ slicing results for the 6th May 2010. Overplotted on this is the CALIPSO track. (b) The optimal estimation scheme heights. (c) The ash mass obtained with the optimal estimation scheme. (d) The CALIOP backscatter plot, with the $CO_2$ slicing results and the optimal estimation scheme heights plotted on top. Indicated on the top left hand side of the plot is the time of the CALIOP overpass. The dashed line indicates the height of the tropopause. (e) Plot of the ash mass corresponding to pixels shown in (d).



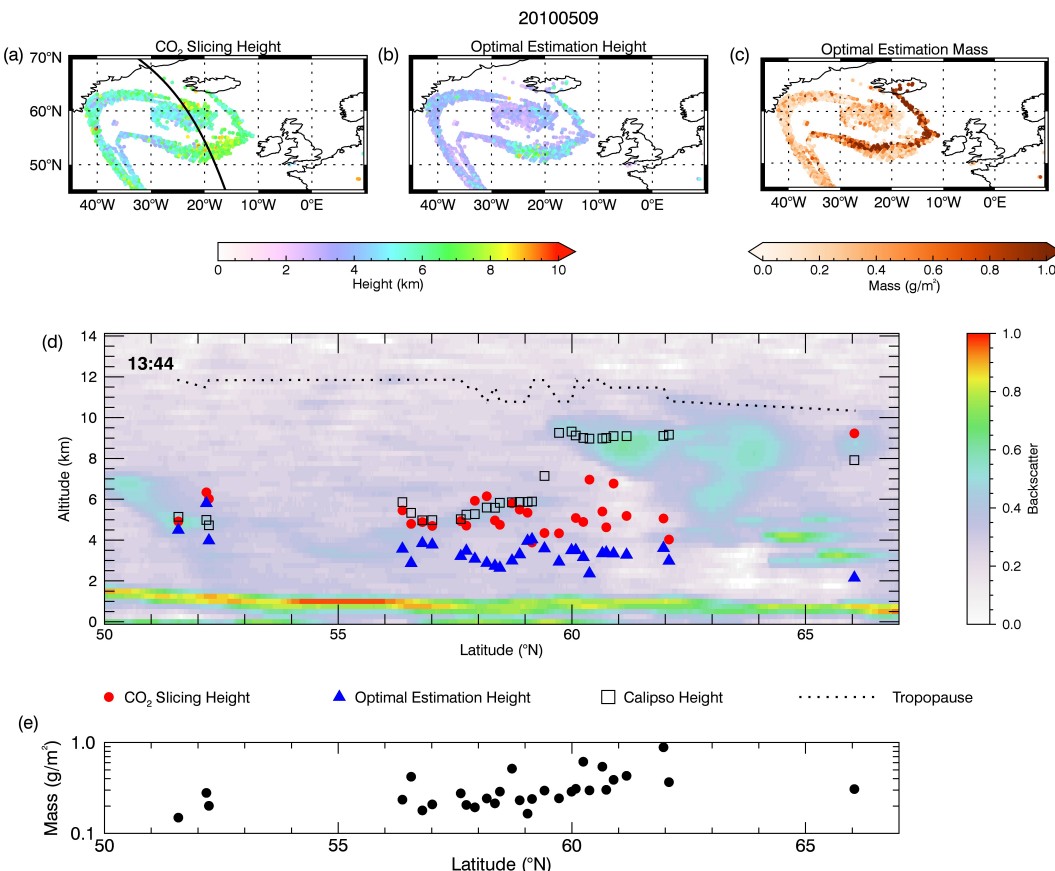

**Figure 11.** Same as figure 10 for $9^{th}$ May 2010.



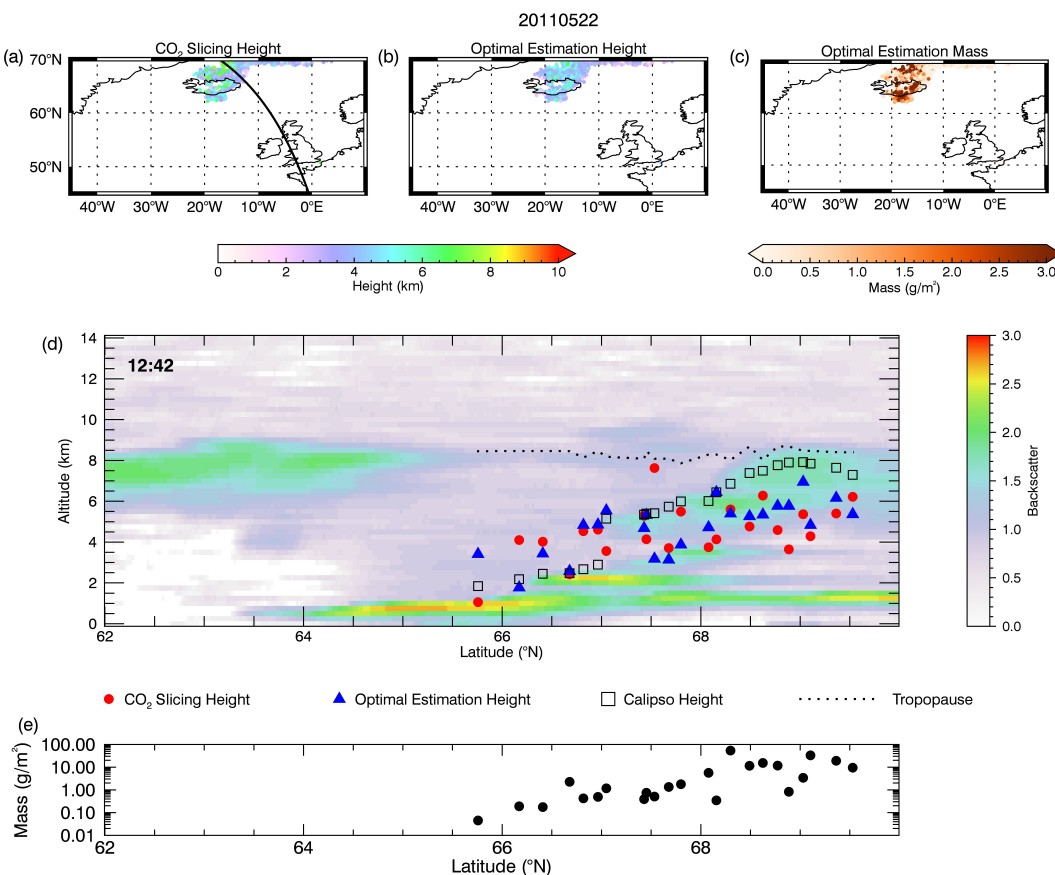

**Figure 12.** Same as figure 10 for $22^{nd}$ May 2011.



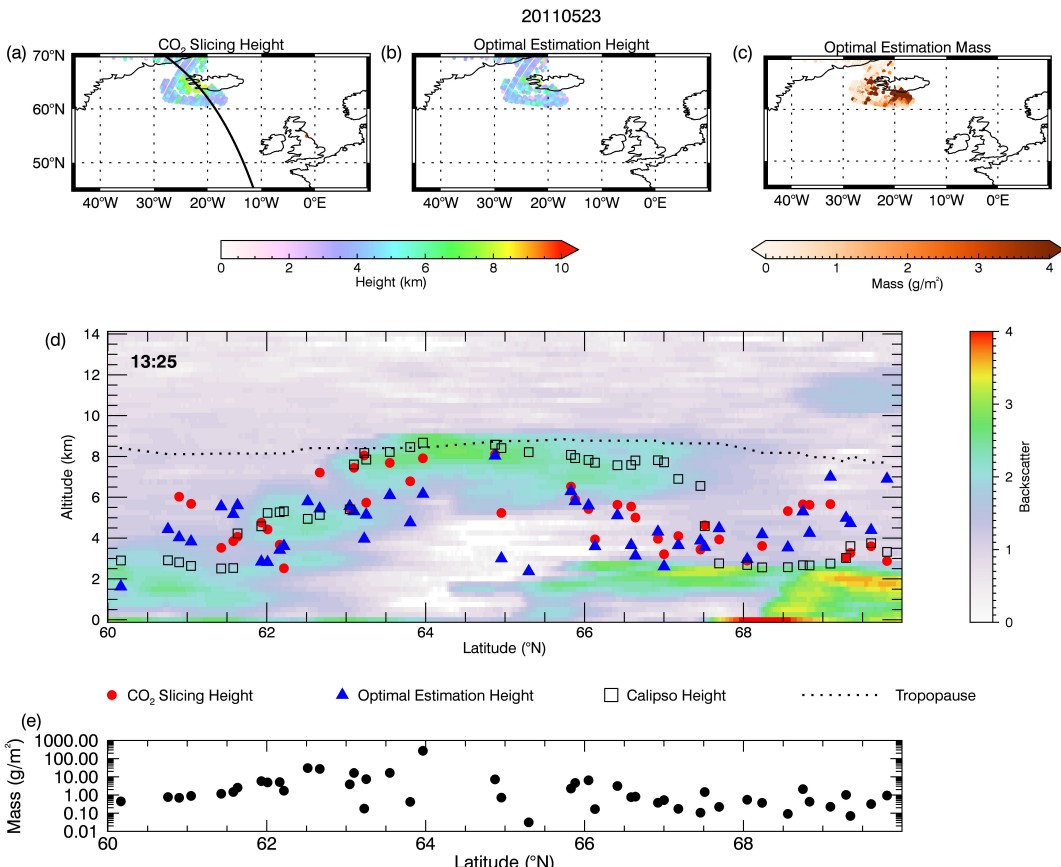

**Figure 13.** Same as figure 10 for $23^{rd}$ May 2011.





**Table 1.** A summary of some of the existing methods for determining the height of volcanic ash clouds. Summaries can be found in Oppenheimer (1998); Prata and Grant (2001a, b); Zakšek et al. (2013)

| Method | Description | Examples in literature |
|---|---|---|
| *Ground based methods* | | |
| **Infrared camera** | Infrared cameras measure the heat radiated off the ash cloud. This means the plume can be distinguished from its surroundings. The top of the plume can be identified and the height calculated by counting the number of pixels between the plume top and a reference point. | Patrick (2007); Sahetapy-Engel and Harris (2009); Webb et al. (2014); Bombrun et al. (2018) |
| **Radar** | A pulse of radio energy is emitted from a transmitter. This is reflected back off clouds (meteorological or ash). This echo can be used to determine the cloud height. | Lacasse et al. (2004); Arason et al. (2011); Petersen et al. (2012) |
| *Multiple platforms* | | |
| **LiDAR** | LiDAR is an active sensor which can be used on the ground as well as on aircraft or satellite platforms. The backscatter returned to the instrument can be used to infer the height of multiple cloud layers (including different types of cloud and ash). This is commonly used for validation of other methods. | Ansmann et al. (2010); Marenco et al. (2011); Winker et al. (2012); Vernier et al. (2013); Balis et al. (2016) |
| *Satellite techniques* | | |
| **Stereo view** | This method requires two instruments viewing the cloud at the same time or a single instrument with two viewing angles (i.e. nadir and forward viewing). The resulting parallax can be used to determine the cloud height. | Prata and Turner (1997); Zakšek et al. (2013) |
| **Cloud shadow** | The shadow cast by clouds can be identified in visible satellite imagery. Combined with knowledge of the satellite viewing angle and the position of the sun, this can be used to find the height of the cloud layer. Alternatively multiple images including the cloud's shadow can be used. | Holasek et al. (1996); Prata and Grant (2001b) |
| **Cloud top temperature** | The cloud top temperature measured by an infrared instrument (usually at 11 $\mu$m) is compared against a temperature profile (e.g. radiosonde or weather model) to obtain the height. | Holasek et al. (1996) |
| **Backward trajectory Modelling** | Method uses the vertical wind directions and backwards trajectory modelling to get vertical distribution of ash. This can then be used to obtain the flux. | Eckhardt et al. (2008)[1],Stohl et al. (2009)[2],Kristiansen et al. (2010)[1],Stohl et al. (2011)[2], Pardini et al. (2017, 2018)[1] |
| **Radiance fitting** | Spectra are forward modelled given certain atmospheric parameters. These spectra are compared against those measured by the instrument and this is used to determine the altitude | Ventress et al. (2016); Zhu et al. (2017) |

[1]Example using $SO_2$ not ash

[2]Example using hydrofluorocarbons and hydrochlorofluorocarbon



**Table 2.** A summary of some of the previous applications of the $CO_2$ slicing technique.

| Instrument | Platform type | Examples |
| --- | --- | --- |
| AIRS | Satellite | Pangaud et al. (2009) |
| GOSAT | Satellite | Someya et al. (2016) |
| IASI | Satellite | Arriaga (2007) |
| ITPR | Satellite | Smith and Platt (1978) |
| MODIS | Satellite | Menzel et al. (1992); Richards (2006)*; Tupper et al. (2007)*; Menzel et al. (2008) |
| MODIS MAS | Airborne | Frey et al. (1999) |
| S-HIRS | Airborne | Holz et al. (2006) |
| VAS | Satellite | Menzel et al. (1983); Wylie and Menzel (1989) |

AIRS- Atmospheric Infrared Sounder

GOSAT- The Greenhouse Gases Observing Satellite

IASI- The Infrared Atmospheric Sounding Interferometer

ITPR- Infrared Temperature Profile Radiometer

MODIS- Moderate Resolution Imaging Spectroradiometer

MODIS MAS- Moderate Resolution Imaging Spectroradiometer Airborne simulator

VAS- Visible Infrared Spin-Scan Visible Radiometer Atmospheric Sounder

*Studies applied to ash



**Table 3.** The channel ranges selected for the final application of the $CO_2$ slicing technique. In total 57 channels are used. Following Arriaga (2007) 900.50 cm$^{-1}$ is used as the window channel used to calculate the effective emissivity

| Step | $CO_2$ Channel Range (inclusive) | Reference Channel | Peak Sensitivity Range | Number of Channels |
|---|---|---|---|---|
| 1 | 700 - 703.5 | 715 | 110.25 - 314.00 | 15 |
| 3 | 706 - 710.5 | 715 | 328.75 - 478.00 | 19 |
| 4 | 713 - 713.5 | 725 | 442.00 - 496.75 | 3 |
| 5 | 718.25 - 719.5 | 728 | 133.75 - 441.75 | 6 |
| 6 | 720.5 - 721.5 | 728 | 21.00 - 496.50 | 5 |
| 7 | 729.75 - 731.75 | 735 | 535.25 - 639.25 | 9 |



**Table 4.** Summary of the percentage of accepted retrievals and the RMSE describing the difference between the true (simulated) and retrieved values

| Atmosphere | No quality control | | With quality Control | |
|---|---|---|---|---|
| | **RMSE (m)** | **Success Percentage** | **RMSE (m)** | **Success Percentage** |
| **RTTOV Standard** | 706 | 91 | 424 | 64 |
| **Mid-Latitude Day** | 635 | 100 | 282 | 84 |
| **Mid-Latitude Night** | 635 | 100 | 282 | 84 |
| **Tropical Day** | 1483 | 100 | 141 | 72 |
| **Polar Summer** | 1271 | 95 | 777 | 29 |
| **Polar Winter** | 565 | 100 | 1553 | 97 |
| **All** | 988 | 97.7 | 777 | 71.9 |





**Table 5.** Statistics describing the comparison of the $CO_2$ slicing and optimal estimation scheme against the heights obtained with CALIOP

| Volcano | $CO_2$ slicing | | | Optimal Estimation | | |
|---|---|---|---|---|---|---|
| | Number of pixels | Correlation Coefficient | RSME (km) | Number of pixels | Correlation Coefficient | RSME (km) |
| Eyjafjallajökull | 53 | 0.2 | 2.2 | 67 | -0.1 | 3.2 |
| Grimsvötn | 65 | 0.5 | 2.1 | 69 | 0.3 | 2.4 |
| All | 118 | 0.4 | 2.2 | 136 | 0.1 | 2.8 |





**Figure A1.** Simulation results for an RTTOV default atmosphere. The top two line shows the maximum difference between the true (simulated) and retrieved pressures grouped into the different pressure levels. Each level consists of ash optical depths ranging between 5 and 15 and effective radius between 5 and 10 $\mu$m. The bottom two lines show the percentage of accepted retrievals (i.e. the percentage of cases where there is an intersection between Eq. 1 and 2, and where all quality control criteria are met).





**Figure A2.** Same as figure A1 for a mid-latitude day atmosphere



**Figure A3.** Same as figure A1 for a mid-latitude night atmosphere

**Figure A4.** Same as figure A1 for a tropical atmosphere





**Figure A5.** Same as figure A1 for a polar summer atmosphere





**Figure A6.** Same as figure A1 for a polar winter atmosphere





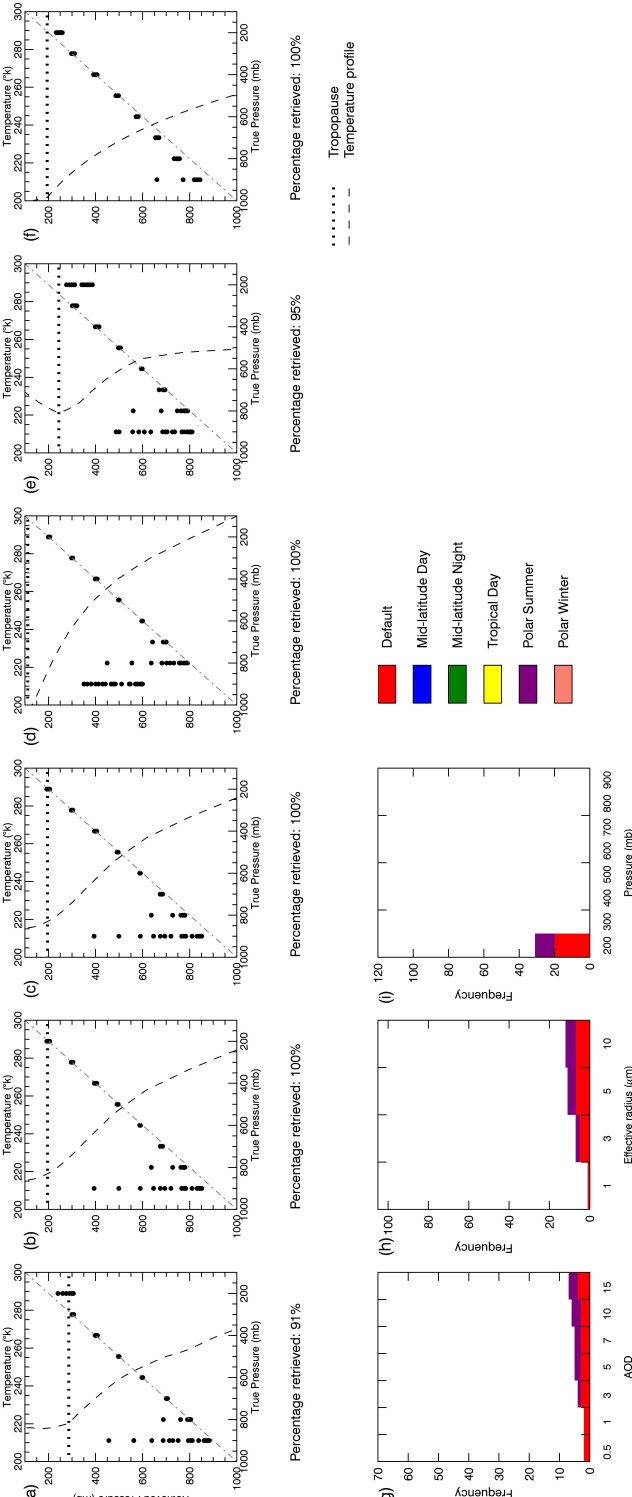

**Figure A7.** Final CO$_2$ slicing pressure results for RTTOV simulated ash spectra (a total of 224 spectra per atmosphere). The plots show the true (simulated) pressure plotted against the CO$_2$ slicing retrieved value for the six different atmospheres. (a) RTTOV default atmosphere (high latitude), (b) Mid-latitude day, (c) Mid-latitude night, (d) Tropical day, (e) Polar summer (f) Polar winter. No quality control has been applied. In this case, the simulated spectra include the following ash properties: ash optical depth ranging between 0.5 and 15, ash effective radius ranging between 1 and 10 $\mu m$ and pressure values between 200 and 900 mb. Below each plot is a value indicating the percentage of successful retrievals. (g) The frequency of ash optical depths for which the CO$_2$ slicing technique was unable to return a height value. (h) Same as (g) for the effective radius. (i) Same as (g) for the ash cloud pressure.