# Peer review of "An adaptation of the CO2 slicing technique for the Infrared Atmospheric Sounding Interferometer to obtain the height of tropospheric volcanic ash clouds"

_Atmospheric Measurement Techniques, 2018_

## Referee Comment (RC1) · Anonymous Referee #1 · 25 Jan 2019

General Comments:

The authors adapt the well-known CO2 slicing technique to obtain the height of tropospheric volcanic ash clouds. They perform simulations to select optimal channels for determining the ash layer height. Subsequently, they employ the technique on IASI spectra obtained during the Eyjafjallajökull and Grimsvötn eruptions, and compare the results against optimal estimation and CALIOP retrievals. These comparisons show that the CO2 slicing technique outperforms optimal estimation. Considering the computational speed of this algorithm, the authors suggest that the technique could be

used to obtain a first approximation of ash height, which could help with hazard mitigation, especially for aircraft navigation. It could also be used to obtain good a priori data for optimal estimation retrievals. The work is scientifically sound and socially relevant. I would recommend publication of the manuscript after the changes listed below are implemented.

Specific Comments:

The words "meteorological cloud" is used a lot. It is not obvious what that term means. Some explanation should be given to introduce it.

Line 12, page 3: "Assuming an atmosphere which is decreasing in temperature with height": This is not a realistic assumption since it is true only for the troposphere. There is some discussion on this in the last paragraph of section 2 but it should probably be moved here.

Do the simulated spectra cover the range of atmospheric conditions expected over the globe? The authors use six different atmospheres: high latitude, mid-latitude day and night, tropical daytime and polar summer and winter. What about tropical night, summer and winter in the tropics and mid-latitudes, etc?

Lines 6-7, page 6: Need a reference(s) showing that those are appropriate values for ash cloud properties.

How is the weighting function $w = d(tau)/d(lnp)$ computed?

Fig. and Figure are both used. Choose one and be consistent.

It is surprising that a technique using just a few channels (CO2 slicing) outperforms one that uses many more channels and retrieves several parameters self-consistently using radiative transfer simulations and iterative fits to spectra (OE). The authors suggest that this may be due to the OE retrievals being strongly influenced by the height a priori. This may be so. However, this suggests that the measurements do not have much information on ash height (otherwise, the prior should not strongly affect the retrievals).

If that is the case, how does the CO2 slicing obtain a better retrieval? A qualitative discussion of the difference between the two techniques is in order (not just that the results are different, but why they are different).

Technical Comments:

Line 7, page 2: which can result is -> which can result in

Line 10, page 3: need reference for RTTOV

Line 28, page 4: remove "That"

Line 30, page 4: "The second" -> "The third"

Line 8, page 5: dependant -> dependent

Line 1, page 9: demonstrated -> demonstrates

Line 23, page 9: including the CO2 slicing technique -> including those obtained using the CO2 slicing technique

Figure 3 caption: lines of the plot -> rows of the plot

Figure 6 caption: The plots show the true (simulated) pressure plotted against the CO2 slicing retrieved value for the six different atmospheres. -> Panels (a)-(f) show the true (simulated) pressure plotted against the CO2 slicing retrieved value for the six different atmospheres.

Figure 8 caption: The authors should note that the maroon distribution represents CALIOP retrievals.

---

## Referee Comment (RC2) · Anonymous Referee #3 · 5 Feb 2019

The paper describes the possibility to use the CO2 slicing established technique to retrieve the top height of tropospheric volcanic ash clouds. The topic is really interesting because the plume altitude is an input of many retrieval scheme and a good estimation of the height is essential to obtain accurate values of ash properties (aerosol optical depth, effective radius, abundance and concentration). The authors adapted and applied the CO2 slicing method to IASI multispectral measurements. Simulated ash spectra were used to select the most useful channels and to validate the procedure, showing good results. Then the technique was applied to real ash spectra from 2010-

2011 Eyjafjallajokull and Grimsvotn eruptions and compared with the results obtained from an optimal estimation scheme and with CALIOP measurements. The manuscript is well written and clear, the work is scientifically accurate and correct. I would recommend publication with some minor changes listed below.

Specific Comments:

- In the description of simulated ash spectra (section 4.1, page 6, lines 6-7), are specified the AOD, Effective Radius and Cloud Heights ranges used, but no reference about the ash type considered, that is the aerosol optical properties (extinction and absorption coefficient, asymmetry parameter, ecc . . .) used in simulation. Which ash type was used? Andesite? Obsidian? Pumice? Other?

- Why you use different AOD and Effective Radius ranges for channel selection (section 4.1) and simulation results (section 4.2)? AOD=5-15 and Ref=5-10 micron for channel selection, AOD=0.5-15 and Ref=1-10 micron for simulation results. Can you explain better?

- I think it would be very interesting to evaluate the heights obtained from CO2 slicing as a function of AOD, Ref and cloud top pressures. In fig. 6 (g-i) are shown the frequency for which the CO2 slicing was unable to return a height value. I suggest to insert 3 similar panel to show the frequency for which the CO2 slicing returns a good value (for example a value that differ from the truth max +/- 500 meters or +/- 1 km). In this way we could better understand in which conditions the CO2 slicing is applicable and reliable.

Technical Comments:

- Page 4, line 1: transmittance is not radiance, so it can't be "emitted" . . .. I suggest to replace the sentence with: "the atmospheric transmittance at channel v of the layer between the pressure level p and the instrument (top of atmosphere)".

- Page 5, line 3: "with" instead of "which" ?

- Page 10, line 6: here you said "4 days" for Grimsvotn eruptions, while in fig. 7 the days are only 3 (20110521 PM, 20110522 AM, 20110522 PM, 20110523 AM).

- Figure 2: the y-label is "Pressure (mbar)" not "Altitude (km)".

- Figure 3: the x-label of the last two lines is missing ("CO2 Wavenumber (cm-1)").

- Figure 4: the same as above.

---

## Referee Comment (RC3) · Anonymous Referee #4 · 12 Feb 2019

General comments: This paper described a technique of the CO2 slicing for the estimation of ash clouds height from satellite infrared sounder data. As the author mentioned, the CO2 slicing is an established method. However, accuracy of the estimation strongly depends on the selected channels and their combination, as well as the adopted atmospheric profiles. The author investigated the dependence from the results of many radiative transfer calculations. Furthermore, it is found that the CO2 slicing technique gives better estimates than Optimal Estimation (OE). This paper is well written and including useful and important suggestion to the researchers of volcanic ash clouds.

In this reason, the reviewer concluded that this paper is suitable for AMT publication. Some minor comments are listed below.

Specific comments:

Page 3 line 10: Add version of RTTOV.

Page 3 line 28: The expression "L_obs $\nu$_1" should be replaced to "L_obs ($\nu$_1 )".

Page 4 line 7: "w" is used for window channel in Eq. (3) but later it uses for weighting function.

Page 4 line 9: There are no explanation for L_cld ($\nu$) in the text. Is it the same as L_obs ($\nu$)?

Section 4: In the approach of CO2 slicing of this paper, contribution of meteorological clouds seems to be omitted in radiance calculations. If so, it should be mentioned in the text.

Page 6 line 5-6: Add reference for the atmospheric profiles.

Page 6 line 6-7: Add the applied ash model of refractive index for ash optical properties. Is it Pollack andesite model?

Page 6 line 13-14: The values or reference for the noise of the instrument should be added.

Page 8 line 28-29: More explanation for the flagged pixel is required. Do you determine the flagged pixels by yourself? What channel and threshold value are used? If the flagged pixels were given from somewhere, the data source should be added.

Page 8 line 29-30: Add description of surface condition (temperature and emissivity) for the calculations of L_clr. Did you use the surface emissivity model in RTTOV?

Page 9, Sec 5.1.1: Detailed explanation for the determination of the a priori ash height in the optimal estimation scheme is needed in the text. It is the important point in the

discussion for the results of comparison between OE and your CO2 slicing.

Figure 2: Label of the ordinate seems wrong. Add values of $\nu\_1$ and $\nu\_2$ in this calculation.

Figure 10-13: There are no discussions for the plots of ash mass (e). Add discussion if these plots are important.

Table 3: What does the number of "step" in table 3 mean? Why step 2 does not exist?

Figure A7: In the caption of Fig. A7, same sentences as those of Fig.6 is not necessary.

---

## Referee Comment (RC4) · Anonymous Referee #2 · 15 Feb 2019

General comments:

In general, I thought this was a really good paper, extremely well-written, and presents some nice, potentially useful results. My only major concern is that I feel that there is a potentially useful scientific discussion to be had as to why the CO2 slicing approach gives better results than the full OE retrieval - and this discussion wasn't really explored to any significant extent in the present manuscript.

Major comments:
1. Abstract and elsewhere: The statement in the abstract reads "Overall, the CO2 slicing tool performs better than the optimal estimation scheme", which stopped me in my tracks! In common with the report of Referee #1, I was surprised by this – why indeed should the CO2 slicing approach (which is basically a cut-down version of a full OE retrieval) outperform the OE approach? I feel that this should be explored in some more detail, because this (for me) is the main scientific issue in this manuscript. The authors suggest that the prior height constraint is responsible for the low ash height bias shown by the OE retrievals – to me, this suggests that the prior is clearly not appropriate in this case, or is being given far too much weight in the analysis.

2. Section 4.1, paragraph beginning "Figure 3 demonstrates...": I felt that this paragraph didn't really do justice to the description of Figure 3, and I think it would be good if the reader could be "guided" through the details of this Figure a little bit more. When you say "...the best performing channels...", do you mean CO2 channels, reference channels, or both? When you say "As expected, this shifts from lower wavenumbers at lower pressures to higher wavenumbers closer to the surface", I found it very hard to interpret the intended meaning. Is "this" referring to the "best performing channels"? I really couldn't reconcile this sentence with what I was seeing in Figure 3.

3. Section 5.2, page 10, 28-29: You suggest that "In future applications of the OE scheme, the CO2 slicing results could be used as the a priori". I disagree very strongly with this statement! One CANNOT use as prior information a state which has already been influenced by the measurements themselves. You could use the CO2-slicing solution as the first guess in the OE iterative process, but absolutely not as the prior constraint. There's an equivalent comment in Section 6 (line 16 on page 12).

Minor comments, grammar, typos, suggestions, etc.

1. Section 2, page 4, line 2: "dependant" in this context should, I think, be "dependent"?

2. Section 3, page 5, line 16: Trivial I know, but EUMETSAT usually insist that the satellite name is "Metop" and not "MetOp"!

3. Section 4.1, and elsewhere: I note that you use "mb" for pressure units – I suspect the journal would prefer "hPa".

4. Section 4.1, page 6, line 9: When you use the phrase "is greater than the CO2 channel", in what sense is "greater than" meant in this context? Channel number, wavenumber? Best to be absolutely explicit for clarity.

5. Section 4.2, page 6, line 16: When you say "The top two lines show...", I suggest using the word "rows" instead of "lines".

6. Section 4.2, page 6, lines 26 and 29: You use the phrase "less channels" a couple of times. It should be "fewer channels".

7. Section 4.2, page 7, lines 16-19: It's not made clear in the text why you make the distinction between using narrower ranges for the channel selection work (ash optical depths ranging between 5 and 15, ash effective radius between 5 and 10 microns) than for the simulated retrieval work (ash optical depths ranging between 0.5 and 15, ash effective radius between 1 and 10 microns). It would be good for you to be explicit as to exactly why you didn't use "spectra representative of thinner ash clouds" for the channel selection.

8. Section 5, page 8, line 31: "planck" should have upper-case "P".

9. Section 5, page 9, line 33: You say that you have defined the tropopause "as the height at which the temperature profile inverts and has a positive gradient", but it's confusing that the tropopause dashed lines in Figures 6(a-f) have obviously not used this definition!

10. Section 5, page 9, lines 1-2: When you say "Figure 6 demonstrated that the CO2 slicing method performs poorly where the temperature profile steepens significantly", can you clarify exactly what "steepens" means in this context – when dealing with (negative) vertical gradients, it's very easy for the reader to become confused with words such as "steepens"!

11. Section 5.2, page 10, lines 31-32: You say that "Ventress et al. (2016) identified that in some cases the retrieval assumed a lower altitude and a higher ash optical depth in order to fit the spectra". Lower/higher than what? Just needs to be a little clearer.

12. Figure 4: What is the x-axis here? The y-axis is labelled simply as "Wavenumber". Presumably they should both be the same as for Figure 3?

13. Figures 10-13: Where does the retrieved ash mass come from – the OE retrievals? In any case, it's not clear to me exactly why these mass column loadings have been included in the paper, as I don't believe they are ever referred to. What do they add to the paper?

14. Table 3: Needs units! Presumably the "Channel Ranges" etc. refer to wavenumbers (cm^-1)? Are the "Peak Sensitivity Ranges" in mb/hPa?

---

## Author Comment (AC1) · 23 Apr 2019

**Response to Anonymous Referee #1**

I would like to begin by thanking the reviewer for the time they've taken to read and constructively comment on the manuscript titled 'An adaptation of the $CO_2$ slicing technique for the Infrared Atmospheric Sounding Interferometer to obtain the height of tropospheric volcanic ash clouds' (amt-2018-447). We believe that these comments have helped to improve the content and clarity of the manuscript and we hope that you agree.

Below are responses to the comments made. The reviewer's comments are coloured in blue and are in bold font. Our responses are offset from these and in black. Text in italics are relevant passages from the revised text.

Specific Comments

**The words "meteorological cloud" is used a lot. It is not obvious what that term means. Some explanation should be given to introduce it.**

> To avoid confusion, we now use the words "aqueous clouds" and define this when it is mentioned in the introduction:
>
> *'Similar retrieval techniques exist to obtain the cloud height of aqueous clouds (i.e. water/ice clouds not associated with volcanic activity)'*

**Line 12, page 3: "Assuming an atmosphere which is decreasing in temperature with height": This is not a realistic assumption since it is true only for the troposphere. There is some discussion on this in the last paragraph of section 2 but it should probably be moved here.**

> This has been rephrased:
>
> *'In the Earth's troposphere where temperature is decreasing with height, the radiances measured by the instrument are proportional to the transparency of the atmosphere for each channel'.*

**Do the simulated spectra cover the range of atmospheric conditions expected over the globe? The authors use six different atmospheres: high latitude, mid-latitude day and night, tropical daytime and polar summer and winter. What about tropical night, summer and winter in the tropics and mid-latitudes, etc?**

> We believe that the six atmospheres chosen captured sufficient atmospheric variability to demonstrate the applicability of the method. From our results the method is weakest as the atmospheric lapse rate approach zero (polar summer).  As the suggested atmospheres are far from isothermal, we do not believe much would be learned given the computational expense.

**Lines 6-7, page 6: Need a reference(s) showing that those are appropriate values for ash cloud properties.**

> In this study we have used a range simulated ash spectra. For the channel selection we use ash spectra representative of optically thick volcanic ash clouds (AODS: 5-15, ER: 5-10). We then test this on a

wider range of ash properties which represent thinner plumes before applying the technique to real ash scenes. We have expanded the first paragraph in section 4.1 to indicate why we chose these ranges and to emphasise that it is then tested on a wider range of properties including those more representative of thinner ash clouds:

*'IASI has over 300 channels which fall within the $CO_2$ absorption band, and so, to ensure computational efficiency an appropriate subset of these channels must be selected. To do this the $CO_2$ slicing technique was first applied to 384 simulated ash spectra. These are `ideal' test cases which do not include other aerosols or aqueous cloud. These spectra include six different atmospheres: high latitude, mid-latitude day and night, tropical daytime and polar summer and winter (including atmospheric profiles created for MIPAS; Remedios et al., 2007). The spectra were modelled using the refractive indices of samples of volcanic ash from the Eyjafjallajökull eruption in 2010 (Peter, 2010): the main eruption considered in this study. In the future different refractive indices could be used such as those in Prata et al. (2019). A range of ash properties were explored: cloud heights between 200 and 900 hPa (going slightly above the tropopause), ash effective radius between 5 and 10 μm, and ash optical depths between 5 and 15 (referenced at 550 nm). Typically, the effective radius is less than 8 μm for very fine ash (such as in a distal plume) and between 8 and 64 μm for fine ash (Marzano et al. 2018). The range of ash optical depths is highly variable. Ventress et al. (2016) and Balis et al. (2016) recorded ash optical depths of less than 1.2 from dispersed plumes from Eyjafjallajökull in 2010; however much higher values can be expected closer to the volcano or following large explosive eruptions. The effective radius and AOD explored here for the channel selection is in the upper range and above what might be expected: values which may only be true close to the volcanic vent. The spectrum of an optically thin plume is more difficult to differentiate from a clear spectrum commonly leading to the signal ($I_{obs}(v)$-$I_{clr}(v)$) to be within the instrument noise and subsequently will result in no retrieval. A decision was made to select the channels used using idealised optically thick cases, which may only be true close to the vent, for which the plume should be evident in the majority of the $CO_2$ channels. The selected channels are tested on a wider range of AODs and effective radius in section 4.2 including smaller values that are more representative of a disperse plume.'*

**How is the weighting function w = d(tau)/d(lnp) computed?**

We compute the derivative of tau (output from RTTOV) with logarithm of pressure for the $CO_2$ channel using the IDL deriv routine. This gives weighting function profiles such as those seen in figure 1d.

The use of the weighting function (now k) to obtain a single pressure solution was originally mentioned on page 7, line 16. We've expanded this sentence to explain what this means and where the value for τ comes from:

*'... where $p_c$ is the pressure retrieved for channels v and k refers to the weighting function based on the derivative of atmospheric transmittance computed for each pressure level with RTTOV with respect to the log of atmospheric pressure (dτ[v,p]/dlnp).'*

**Fig. and Figure are both used. Choose one and be consistent.**

We have followed the convention outlined by AMT in the Manuscript preparation guidelines. We have used Fig. where it appears in the running text and Figure at the start of a sentence.

**It is surprising that a technique using just a few channels (CO$_2$ slicing) outperforms one that uses many more channels and retrieves several parameters self-consistently using radiative transfer simulations and iterative fits to spectra (OE). The authors suggest that this may be due to the OE retrievals being strongly influenced by the height a priori. This may be so. However, this suggests that the measurements do not have much information on ash height (otherwise, the prior should not strongly affect the retrievals). If that is the case, how does the CO$_2$ slicing obtain a better retrieval? A qualitative discussion of the difference between the two techniques is in order (not just that the results are different, but why they are different).**

In this study, we have compared the CO$_2$ slicing results against the height output from an optimal estimation scheme, the results of which have been published previously (Ventress *et al.* 2016). This optimal estimation technique uses 105 channels, 14 of which are within the CO$_2$ absorption band. The channels used were not selected for their ability to obtain the ash cloud height and the previous study acknowledged that this is something that could be improved. Where there is not sufficient information about the height within the channels then the output would tend to the prior. Changes could be made to the OE retrieval, such as the inclusion of further channels within the CO$_2$ absorption band and this might improve the results. In this case however, we are comparing our results against the previously published study.

To avoid misleading the reader, we have removed the statement saying that the '*CO$_2$ slicing technique performs better than the OE technique*' (previously **page 1, line 13**) as re-reading this, this might imply that the CO$_2$ slicing method performs better than *any* optimal estimation scheme rather than just the version chosen for comparison.

We have also reworded the discussion of why the output of the two retrievals is different and improved the description of the a priori:

*'By contrast, the OE average heights are less variable: between 3 and 4.25 km throughout the period studied. Some example maps of the OE results are shown in Fig. 10 to 13. The different assumptions and limitations of the two techniques mean that it is not expected that the two retrievals will return the same or even similar values. The optimal estimation scheme uses only 105 channels between 680.75 and 1204.5 cm$^{-1}$ (~8.3 - 14.6 µm) to improve computational efficiency. This includes 14 channels within the CO$_2$ absorption band, only one of which is in common with the CO$_2$ slicing. However, unlike the CO$_2$ slicing method presented here, the channels used by the optimal estimation scheme have not been optimised for retrieving the height of the ash layer. Ventress et al. (2016) noted that the optimal estimation retrieval could be further refined by altering the channels used. For example, channels with more height information could be selected. Similarly, Ventress et al. (2016) suggested that channels could be selected to minimise the effect of the underlying cloud layers following observations that the OE method can underestimate the cloud top height in cases of multiple cloud layers (Ventress et al. 2016). In the current application of the optimal estimation scheme, where there is not sufficient information about the height of the ash layer within the channels used, the retrieval height output will tend to the a priori height which in this case is around 3.5 km. This is potentially the reason for the persistently lower average height shown in Fig. 9 which suggests a strong dependence on the a priori.'*

Technical Comments

**Line 7, page 2: which can result is -> which can result in**

This has been corrected.

**Line 10, page 3: need reference for RTTOV**

This has been added:

*'This has been simulated with the fast radiative transfer model RTTOV (version 9, Saunders et al. 1998) …'*

**Line28, page 4: remove "That"**

Done

**Line 30, page 4: "The second" -> "The third"**

The original passage read: *'(2) the two channels used in Eq. 1 are sufficiently close that the difference in emissivity between them is negligible; (3) in cases where there are multiple layers of cloud, the lower level clouds are ignored. The second is particularly important to consider when the channel pairs are selected.'*

The channel pairs selected must have a negligible emissivity difference – this is referring to the second stated assumption. This paragraph has been restructured to avoid confusion:

*'The $CO_2$ slicing method makes a number of assumptions: (1) the cloud is infinitesimally thin; (2) in cases where there are multiple layers of cloud, the lower level clouds are ignored; (3) the two channels used in Eq. 1 are sufficiently close that the difference in emissivity between them is negligible: this is particularly important to consider when the channel pairs are selected. Multiple cloud layers have previously been identified as a source of error in the $CO_2$ slicing retrieval …'*

**Line 8, page 5: dependant -> dependent**

Done

**Line 1, page 9: demonstrated -> demonstrates**

Done

**Line 23, page 9: including the $CO_2$ slicing technique -> including those obtained using the $CO_2$ slicing technique**

This has been reworded:

*'CALIOP and other LiDAR instruments are commonly used as a tool for the validation of cloud heights, including previous studies with the $CO_2$ slicing technique'*

**Figure 3 caption: lines of the plot -> rows of the plot**

Changed.

**Figure 6 caption: The plots show the true (simulated) pressure plotted against the $CO_2$ slicing retrieved value for the six different atmospheres. -> Panels (a)-(f) show the true (simulated) pressure plotted against the $CO_2$ slicing retrieved value for the six different atmospheres.**

Changed

**Figure 8 caption: The authors should note that the maroon distribution represents CALIOP retrievals**

Panels (a) and (b) of figure 8 show the distribution of the $CO_2$ slicing (red/maroon) and optimal estimation (blue) heights obtained for the Eyjafjallajökull and Grímsvötn eruptions respectively. This is for all the pixels to which the retrieval was applied. Neither shows the distribution of heights obtained from CALIOP.

Panels (c) and (d) show scatterplots comparing the heights obtained with CALIOP (x axis) with those retrieved by the two IASI retrievals (CO2 slicing in red and OE in blue) again for both eruptions.

We've expanded the figure caption to make this easier to understand and added a legend for panels (c) and (d):

[Figure]

*'(a) Distribution of the CO$_2$ slicing and optimal estimation retrieved ash heights for all pixels from the Eyjafjallajökull eruption. (b) Same as (a) for the Grímsvötn eruption. (c) Comparison of the CALIOP heights with those obtained with the CO$_2$ slicing and optimal estimation techniques for a subset of pixels (where measurements fell within 50 km and 2 hours of each other) from the Eyjafjallajökull eruption. (d) Same as (c) for the Grímsvötn eruption. Related statistics can be seen in table 5.'*

---

## Author Comment (AC2) · 23 Apr 2019

**Response to Anonymous Referee #3**

I would like to begin by thanking the reviewer for the time they've taken to read and constructively comment on the manuscript titled 'An adaptation of the $CO_2$ slicing technique for the Infrared Atmospheric Sounding Interferometer to obtain the height of tropospheric volcanic ash clouds' (amt-2018-447). We believe that these comments have helped to improve the content and clarity of the manuscript and we hope that you agree.

Below are responses to the comments made. The reviewer's comments are coloured in blue and are in bold font. Our responses are offset from these and in black. Text in italics are relevant passages from the revised text.

Specific Comments:

**In the description of simulated ash spectra (section 4.1, page 6, lines 6-7), are specified the AOD, Effective Radius and Cloud Heights ranges used, but no reference about the ash type considered, that is the aerosol optical properties (extinction and absorption coefficient, asymmetry parameter, etc . . .) used in simulation. Which ash type was used? Andesite? Obsidian? Pumice? Other?**

An additional line has been included:

*'The refractive index used in this study is from measurements made of ash from the Eyjafjallajökull eruption (Peters, 2010): the main eruption considering in this study. In the future different refractive indices could be used such as those in Prata et al. (2019).'*

**Why you use different AOD and Effective Radius ranges for channel selection (section 4.1) and simulation results (section 4.2)? AOD=5-15 and Ref=5-10 micron for channel selection, AOD=0.5-15 and Ref=1-10 micron for simulation results. Can you explain better?**

**Following here the answer to the same question posed by referee #1:**

In this study we have used a range simulated ash spectra. For the channel selection we use ash spectra representative of optically thick volcanic ash clouds (AODS: 5-15, ER: 5-10). We then test this on a wider range of ash properties which represent thinner plumes before applying the technique to real ash scenes. We have expanded the first paragraph in section 4.1 to indicate why we chose these ranges and to emphasise that it is then tested on a wider range of properties including those more representative of thinner ash clouds:

 *'IASI has over 300 channels which fall within the $CO_2$ absorption band, and so, to ensure computational efficiency an appropriate subset of these channels must be selected. To do this the $CO_2$ slicing technique was first applied to 384 simulated ash spectra. These are `ideal' test cases which do not include other aerosols or aqueous cloud. These spectra include six different atmospheres: high latitude, mid-latitude day and night, tropical daytime and polar summer and winter (including atmospheric profiles created for MIPAS; Remedios et al., 2007). The spectra were modelled using the refractive indices of samples of volcanic ash from the Eyjafjallajökull eruption in 2010 (Peter, 2010): the main eruption considered in this study. In the future different refractive indices could be used such as those in Prata et al. (2019). A range*

*of ash properties were explored: cloud heights between 200 and 900 hPa (going slightly above the tropopause), ash effective radius between 5 and 10 µm, and ash optical depths between 5 and 15 (referenced at 550 nm). Typically, the effective radius is less than 8 µm for very fine ash (such as in a distal plume) and between 8 and 64 µm for fine ash (Marzano et al. 2018). The range of ash optical depths is highly variable. Ventress et al. (2016) and Balis et al. (2016) recorded ash optical depths of less than 1.2 from dispersed plumes from Eyjafjallajökull in 2010; however much higher values can be expected closer to the volcano or following large explosive eruptions. The effective radius and AOD explored here for the channel selection is in the upper range and above what might be expected: values which may only be true close to the volcanic vent. The spectrum of an optically thin plume is more difficult to differentiate from a clear spectrum commonly leading to the signal ($I_{obs}(v)-I_{clr}(v)$) to be within the instrument noise and subsequently will result in no retrieval. A decision was made to select the channels used using idealised optically thick cases, which may only be true close to the vent, for which the plume should be evident in the majority of the $CO_2$ channels. The selected channels are tested on a wider range of AODs and effective radius in section 4.2 including smaller values that are more representative of a disperse plume.'*

**I think it would be very interesting to evaluate the heights obtained from $CO_2$ slicing as a function of AOD, Ref and cloud top pressures. In fig. 6 (g-i) are shown the frequency for which the $CO_2$ slicing was unable to return a height value. I suggest to insert 3 similar panel to show the frequency for which the $CO_2$ slicing returns a good value (for example a value that differ from the truth max +/- 500 meters or +/- 1 km). In this way we could better understand in which conditions the $CO_2$ slicing is applicable and reliable.**

As suggested, three additional panels were added to figure 6 which is shown below. These show the frequency of cases where the difference between the simulated and retrieved values are less than 0.5 km. These plots demonstrate that the $CO_2$ slicing technique performs slightly less well for cases with smaller ash optical depths and effective radius. It also supports the previous observation that the $CO_2$ slicing technique does not perform as well at the pressure extremes tested.

We have included the updated plot in the manuscript and added a few lines in section 4.2:

*'[about failed aods and er] …. This observation is supported by figure 6j-k which shows the number of cases where the difference between the simulated and retrieved pressure is less than 0.5 km: which is slightly lower for a smaller effective radius and ash optical depth.'*

*'[about failed extreme pressures] …. The majority of failed cases are shown to be at the pressure extremes, Fig. 6i. Similarly, Fig. 6l indicates that there are fewer cases where the pressure difference between the simulated and retrieved pressures are less than 0.5 km at these pressures.'*

[Figure]

*Figure 6 caption: Final CO₂ slicing pressure results for RTTOV simulated ash spectra (a total of 224 spectra per atmosphere). Panels (a)-(f) show the true (simulated) pressure plotted against the CO₂ slicing retrieved value for the six different atmospheres. (a) RTTOV default atmosphere (high latitude), (b) Mid-latitude day, (c) Mid-latitude night, (d) Tropical day, (e) Polar summer (f) Polar winter. In this case, the simulated spectra include the following ash properties: ash optical depth ranging between 0.5 and 15, ash effective radius ranging between 1 and 10 µm and pressure values between 200 and 900 hPa. Below each plot is a value indicating the percentage of successful retrievals (where a height value can be obtained and all quality control conditions have been met). (g) The frequency of ash optical depths for which the CO₂ slicing technique was unable to return a height value. (h) Same as (g) for the effective radius. (i) Same as (g) for the ash cloud pressure. (j) The frequency of ash optical depths for which the difference between the simulation and CO₂ slicing height is less than 0.5 km. (k) The same as (j) for effective radius. (l) The same as (j) for ash cloud pressure. Related statistics can be seen in table 4. The equivalent plot, where the values which have not met the quality control conditions has been included in the appendix, figure A7.*

Technical Comments

**Page 4, line 1: transmittance is not radiance, so it can't be "emitted" . . .. I suggest to replace the sentence with: "the atmospheric transmittance at channel v of the layer between the pressure level p and the instrument (top of atmosphere)".**

    Done

**Page 5, line 3: "with" instead of "which" ?**

We've changed which to that. The line now reads: *'The effect of surface emissivity is expected to be minimal as channels within the* $CO_2$ *absorption band have weighting functions that peak above the surface, as shown in Fig. 1d.'*

**Page 10, line 6: here you said "4 days" for Grímsvötn eruptions, while in fig. 7 the days are only 3 (20110521 PM, 20110522 AM, 20110522 PM, 20110523 AM).**

Corrected.

**Figure 2: the y-label is "Pressure (mbar)" not "Altitude (km)".**

Updated to hPa.

**Figure 3: the x-label of the last two lines is missing ("$CO_2$ Wavenumber (cm-1)").**

This has been corrected.

**Figure 4: the same as above.**

This has been corrected.

---

## Author Comment (AC3) · 23 Apr 2019

**Responses to Referee #4**

I would like to begin by thanking the reviewer for the time they've taken to read and constructively comment on the manuscript titled 'An adaptation of the $CO_2$ slicing technique for the Infrared Atmospheric Sounding Interferometer to obtain the height of tropospheric volcanic ash clouds' (amt-2018-447). We believe that these comments have helped to improve the content and clarity of the manuscript and we hope that you agree.

Below are responses to the comments made. The reviewer's comments are coloured in blue and are in bold font. Our responses are offset from these and in black. Text in italics are relevant passages from the revised text.

Specific Comments

**Page 3 line 10: Add version of RTTOV.**

This now reads:

*'This has been simulated with the fast radiative transfer model RTTOV (version 9, Saunders* et al. *1998) and replicates what would be observed with IASI given specified atmospheric conditions'*

**Page 3 line 28: The expression "L_obs v_1" should be replaced to "L_obs (v_1 )".**

Done

**Page 4 line 7: "w" is used for window channel in Eq. (3) but later it uses for weighting function.**

The weighting function has been changed to k

**Page 4 line 9: There are no explanation for L_cld (v) in the text. Is it the same as L_obs (v)?**

Where it occurs $L_{cld}(v)$ has been replaced with $L_{obs}(v)$.

**Section 4: In the approach of $CO_2$ slicing of this paper, contribution of meteorological clouds seems to be omitted in radiance calculations. If so, it should be mentioned in the text.**

We've added an additional line to clarify:

*'To do this, the* $CO_2$ *slicing technique was first applied to 384 simulated ash spectra. These are 'ideal' test cases, which do not include other aerosols or aqueous cloud.'*

**Page 6 line 5-6: Add reference for the atmospheric profiles.**

A reference has been added for the MIPAS atmospheric profiles:

*'These spectra include six different atmospheres: high latitude, mid-latitude day and night, tropical daytime and polar summer and winter (including atmospheric profiles created for MIPAS; Remedios et al. 2007)'*

The reference for these profiles:  https://www.atmos-chem-phys-discuss.net/7/9973/2007/

**Page 6 line 6-7: Add the applied ash model of refractive index for ash optical properties. Is it Pollack andesite model?**

An additional line has been included:

*'The refractive index used in this study is from measurements made of ash from the Eyjafjallajökull eruption (Peters, 2010): the main eruption considered in this study. In the future different refractive indices could be used such as those in Prata et al. (2019).'*

**Page 6 line 13-14: The values or reference for the noise of the instrument should be added.**

We've added this:

*'(1) $L_{obs}v_1$-$L_{clr}v_1$ must be greater than the noise of the instrument at channel $v_1$ ($CO_2$ channel); within the $CO_2$ absorption band the noise of the IASI instruments is between $2.55 \times 10^{-8}$ and $3.77 \times 10^{-8}$ $W/(cm^2.sr.cm^{-1})$)'*

**Page 8 line 28-29: More explanation for the flagged pixel is required. Do you determine the flagged pixels by yourself? What channel and threshold value are used? If the flagged pixels were given from somewhere, the data source should be added.**

More detail has been added to describe the method used to flag the ash pixels prior to the application of the $CO_2$ slicing and optimal estimation techniques:

*'In this application of the retrieval, it has only been applied to pixels which are flagged as containing volcanic ash by a linear ash retrieval developed for IASI (Ventress et al. 2016; Sears et al. 2013): following the method developed for $SO_2$ by Walker et al. 2012). This method compares each IASI spectra against a covariance matrix formed from pixels which contain no volcanic ash thereby representing the spectral variability associated with interfering gas species or clouds, and also the instrument noise. A least squares fit is performed for three ash altitudes (400, 600 and 800 hPa) to retrieve a value for ash optical depth. A pixel is then flagged if it exceeds a threshold at any height. As $SO_2$ can, with caution, be used as a proxy for volcanic ash (Carn et al., 2009; Thomas and Prata, 2011) the retrieval has also been run for pixels flagged for $SO_2$ using the same approach (Walker et al. 2011, 2012; Carboni et al. 2012, 2016).'*

**Page 8 line 29-30: Add description of surface condition (temperature and emissivity) for the calculations of L_clr. Did you use the surface emissivity model in RTTOV?**

We've added a few lines to explain this:

*'For the $CO_2$ slicing values for $L_{clr}$ were obtained using the radiative transfer model RTTOV using the ECMWF atmospheric profile as an input and using the default ocean emissivity within RTTOV. The effect of surface emissivity is thought to be minimal as for the channels used the weighting functions peak above the surface, Fig. 1d.'*

**Page 9, Sec 5.1.1: Detailed explanation for the determination of the a priori ash height in the optimal estimation scheme is needed in the text. It is the important point in the discussion for the results of comparison between OE and your $CO_2$ slicing.**

We have expanded the discussion of the OE scheme within section 5.2 where the results of the OE and $CO_2$ slicing techniques are compared. Within this we give more detail on why the $CO_2$ slicing performs better in these cases and why the OE is affected by the a priori used.

**Following here the answer to a similar question posed by referee #1:**

In this study, we have compared the $CO_2$ slicing results against the height output from an optimal estimation scheme, the results of which have been published previously (Ventress *et al.* 2016). This optimal estimation technique uses 105 channels, 14 of which are within the $CO_2$ absorption band. The channels used were not selected for their ability to obtain the ash cloud height and the previous study acknowledged that this is something that could be improved. Where there is not sufficient information about the height within the channels then the output would tend to the prior. Changes could be made to the OE retrieval, such as the inclusion of further channels within the $CO_2$ absorption band and this might improve the results. In this case however, we are comparing our results against the previously published study.

To avoid misleading the reader, we have removed the statement saying that '*the $CO_2$ slicing technique performs better than the OE technique*' (previously **page 1, line 13**) as re-reading this, this might imply that the $CO_2$ slicing method performs better than *any* optimal estimation scheme rather than just the version chosen for comparison.

We have also reworded the discussion of why the output of the two retrievals is different and improved the description of the a priori:

*'By contrast, the OE average heights are less variable: between 3 and 4.25 km throughout the period studied. Some example maps of the OE results are shown in Fig. 10 to 13. The different assumptions and limitations of the two techniques mean that it is not expected that the two retrievals will return the same or even similar values. The optimal estimation scheme uses only 105 channels between 680.75 and 1204.5 $cm^{-1}$ (~8.3 - 14.6 µm) to improve computational efficiency. This includes 14 channels within the $CO_2$ absorption band, only one of which is in common with the $CO_2$ slicing. However, unlike the $CO_2$ slicing method presented here, the channels used by the optimal estimation scheme have not been optimised for retrieving the height of the ash layer. Ventress et al. (2016) noted that the optimal estimation retrieval could be further refined by altering the channels used. For example, channels with more height information could be selected. Similarly, Ventress et al. (2016) suggested that channels could be selected to minimise the effect of the underlying cloud layers following observations that the OE method can underestimate the cloud top height in cases of multiple cloud layers (Ventress et al. 2016). In the current application of the optimal estimation scheme, where there is not sufficient information about the height of the ash layer within the channels used, the retrieval height output will tend to the a priori height which in this case is around 3.5 km. This is potentially the reason for the persistently lower average height shown in Fig. 9 which suggests a strong dependence on the a priori.'*

**Figure 2: Label of the ordinate seems wrong. Add values of v_1 and v_2 in this calculation.**

Figure 2 has been updated to show pressure on the y axis. We've added the following line to the caption to indicate which channels are used:

*'In this example $v_1$ and $v_2$ are at 715 cm$^{-1}$ and 725 cm$^{-1}$ respectively.'*

**Figure 10-13: There are no discussions for the plots of ash mass (e). Add discussion if these plots are important.**

These panels in figures 10-13 have been removed.

**Table 3: What does the number of "step" in table 3 mean? Why step 2 does not exist?**

This column in table 3 has been removed.

**Figure A7: In the caption of Fig. A7, same sentences as those of Fig.6 is not necessarys**

The caption has been edited and now reads:

*'Same as figure 6 without a quality control applied.'*

---

## Author Comment (AC4) · 23 Apr 2019

**Response to Anonymous Referee #2**

I would like to begin by thanking the reviewer for the time they've taken to read and constructively comment on the manuscript titled 'An adaptation of the $CO_2$ slicing technique for the Infrared Atmospheric Sounding Interferometer to obtain the height of tropospheric volcanic ash clouds' (amt-2018-447). We believe that these comments have helped to improve the content and clarity of the manuscript and we hope that you agree. In particular we have expanded our discussion of the optimal estimation scheme and why we believe the $CO_2$ slicing technique has performed better in the cases studied.

Below are responses to the comments made. The reviewer's comments are coloured in blue and are in bold font. Our responses are offset from these and in black. Text in italics are relevant passages from the revised text.

Major Comments

**1) Abstract and elsewhere: The statement in the abstract reads "Overall, the $CO_2$ slicing tool performs better than the optimal estimation scheme", which stopped me in my tracks! In common with the report of Referee #1, I was surprised by this – why indeed should the $CO_2$ slicing approach (which is basically a cut-down version of a full OE retrieval) outperform the OE approach? I feel that this should be explored in some more detail, because this (for me) is the main scientific issue in this manuscript. The authors suggest that the prior height constraint is responsible for the low ash height bias shown by the OE retrievals – to me, this suggests that the prior is clearly not appropriate in this case, or is being given far too much weight in the analysis.**

> **Following here the answer to the same question posed by referee #1:**
>
> In this study, we have compared the $CO_2$ slicing results against the height output from an optimal estimation scheme, the results of which have been published previously (Ventress *et al.* 2016). This optimal estimation technique uses 105 channels, 14 of which are within the $CO_2$ absorption band. The channels used were not selected for their ability to obtain the ash cloud height and the previous study acknowledged that this is something that could be improved. Where there is not sufficient information about the height within the channels then the output would tend to the prior. Changes could be made to the OE retrieval, such as the inclusion of further channels within the $CO_2$ absorption band and this might improve the results. In this case however, we are comparing our results against the previously published study.
>
> To avoid misleading the reader, we have removed the statement saying that the '*$CO_2$ slicing technique performs better than the OE technique*' (previously **page 1, line 13**) as re-reading this, this might imply that the $CO_2$ slicing method performs better than *any* optimal estimation scheme rather than just the version chosen for comparison.
>
> We have also reworded the discussion of why the output of the two retrievals is different and improved the description of the a priori:
>
> *'By contrast, the OE average heights are less variable: between 3 and 4.25 km throughout the period studied. Some example maps of the OE results are shown in Fig. 10 to 13. The different assumptions and limitations of the two techniques mean that it is not expected that the two retrievals will return the same or even similar values. The optimal estimation scheme uses only 105 channels between 680.75 and 1204.5 cm$^{-1}$ (~8.3 - 14.6 μm) to improve computational efficiency. This includes 14 channels*

*within the $CO_2$ absorption band, only one of which is in common with the $CO_2$ slicing. However, unlike the $CO_2$ slicing method presented here, the channels used by the optimal estimation scheme have not been optimised for retrieving the height of the ash layer. Ventress et al. (2016) noted that the optimal estimation retrieval could be further refined by altering the channels used. For example, channels with more height information could be selected. Similarly, Ventress et al. (2016) suggested that channels could be selected to minimise the effect of the underlying cloud layers following observations that the OE method can underestimate the cloud top height in cases of multiple cloud layers (Ventress et al. 2016). In the current application of the optimal estimation scheme, where there is not sufficient information about the height of the ash layer within the channels used, the retrieval height output will tend to the a priori height which in this case is around 3.5 km. This is potentially the reason for the persistently lower average height shown in Fig. 9 which suggests a strong dependence on the a priori.'*

**2) Section 4.1, paragraph beginning "Figure 3 demonstrates. . .": I felt that this paragraph didn't really do justice to the description of Figure 3, and I think it would be good if the reader could be "guided" through the details of this Figure a little bit more. When you say ". . .the best performing channels. . .", do you mean $CO_2$ channels, reference channels, or both? When you say "As expected, this shifts from lower wavenumbers at lower pressures to higher wavenumbers closer to the surface", I found it very hard to interpret the intended meaning. Is "this" referring to the "best performing channels"? I really couldn't reconcile this sentence with what I was seeing in Figure 3.**

The description of figure 3 was previously on **page 6, lines 24-34**.

We have expanded this paragraph to provide more detail on what is shown in figure 3. We now hopefully guide the reader through this better and demonstrate that the height of the ash layer affects the performance of the different channel combinations:

*'Figure 3 demonstrates that the best performing channel pairs vary depending on the height of the plume. For plumes at lower pressures, the maximum pressure difference between the simulated and retrieved pressures is smaller at lower $CO_2$ wavenumbers. For example, for the plumes simulated at 300 hPa, the maximum pressure difference was lowest (less than 20 hPa) for $CO_2$ channels between 700 and 710 $cm^{-1}$. As the pressure of the ash layer is increased, values are no longer obtained at smaller wavenumbers. For example, for a plume at 500 hPa, solutions are no longer obtained for $CO_2$ channels which are less than 700 $cm^{-1}$: the maximum pressure difference between the true and retrieved values is now smaller for slightly higher wavenumbers. For a plume at 800 hPa the maximum pressure difference is lowest (less than 60 hPa) for $CO_2$ channels between 715 and 720 $cm^{-1}$. This observation reflects what is shown in figure 1b and c: that the channel's peak sensitivity shifts from higher in the atmosphere at lower wavenumbers to close to the surface as higher wavenumbers effecting the best performing channel combination. Notably, at 200 hPa there...'*

**3) Section 5.2, page 10, 28-29: You suggest that "In future applications of the OE scheme, the $CO_2$ slicing results could be used as the a priori". I disagree very strongly with this statement! One CANNOT use as prior information a state which has already been influenced by the measurements themselves. You could use the $CO_2$-slicing solution as the first guess in the OE iterative process, but absolutely not as the prior constraint. There's an equivalent comment in Section 6 (line 16 on page 12).**

The optimal estimation scheme used for comparison in this study uses a total of 105 channels as opposed to the entire spectra to improve computational efficiency. Of these only 14 are within the $CO_2$ band. The optimal estimation scheme only uses one channel which is also used by the $CO_2$ slicing

scheme. If that channel were excluded from the optimal estimation retrieval, then the $CO_2$ slicing result could be used as an a priori as it is not based on the same measurements. This has hopefully been clarified with the addition of further description about the optimal estimation scheme (as described in response to your first comment) and an additional line has been included after the comment about using the $CO_2$ slicing heights as an a priori in section 6:

*'In future applications of the OE scheme, the* $CO_2$ *slicing results could be used as the a priori if the one* $CO_2$ *channel that the two retrievals have in common was removed from the optimal estimation scheme'*

Minor comments, grammar, typos, suggestions, etc.

**1) Section 2, page 4, line 2: "dependant" in this context should, I think, be "dependent"?**

Done

**2) Section 3, page 5, line 16: Trivial I know, but EUMETSAT usually insist that the satellite name is "Metop" and not "MetOp"!**

Done

**3) Section 4.1, and elsewhere: I note that you use "mb" for pressure units – I suspect the journal would prefer "hPa".**

This has been changed throughout.

**4) Section 4.1, page 6, line 9: When you use the phrase "is greater than the $CO_2$ channel", in what sense is "greater than" meant in this context? Channel number, wavenumber? Best to be absolutely explicit for clarity.**

This was referring to wavenumber which has now been stated:

*'The $CO_2$ slicing method was first applied using every channel combination between 660 and 800 cm$^{-1}$, where the reference channel ($v_2$) wavenumber is greater than the $CO_2$ channel ($v_1$) wavenumber.'*

**5) Section 4.2, page 6, line 16: When you say "The top two lines show. . .", I suggest using the word "rows" instead of "lines".**

Done

**6) Section 4.2, page 6, lines 26 and 29: You use the phrase "less channels" a couple of times. It should be "fewer channels".**

Done

**7) Section 4.2, page 7, lines 16-19: It's not made clear in the text why you make the distinction between using narrower ranges for the channel selection work (ash optical depths ranging between 5 and 15, ash effective radius between 5 and 10 microns) than for the simulated retrieval work (ash optical depths ranging between 0.5 and 15, ash effective radius between 1 and 10 microns). It would be good for you to be explicit as to exactly why you didn't use "spectra representative of thinner ash clouds" for the channel selection.**

**Following here the answer to the same question posed by referee #1:**

In this study we have used a range simulated ash spectra. For the channel selection we use ash spectra representative of optically thick volcanic ash clouds (AODS: 5-15, ER: 5-10). We then test this on a wider range of ash properties which represent thinner plumes before applying the technique to real ash scenes. We have expanded the first paragraph in section 4.1 to indicate why we chose these ranges and to emphasise that it is then tested on a wider range of properties including those more representative of thinner ash clouds:

*'IASI has over 300 channels which fall within the $CO_2$ absorption band, and so, to ensure computational efficiency an appropriate subset of these channels must be selected. To do this the $CO_2$ slicing technique was first applied to 384 simulated ash spectra. These are `ideal' test cases which do not include other aerosols or aqueous cloud. These spectra include six different atmospheres: high latitude, mid-latitude day and night, tropical daytime and polar summer and winter (including atmospheric profiles created for MIPAS; Remedios et al., 2007). The spectra were modelled using the refractive indices of samples of volcanic ash from the Eyjafjallajökull eruption in 2010 (Peter, 2010): the main eruption considered in this study. In the future different refractive indices could be used such as those in Prata et al. (2019). A range of ash properties were explored: cloud heights between 200 and 900 hPa (going slightly above the tropopause), ash effective radius between 5 and 10 μm, and ash optical depths between 5 and 15 (referenced at 550 nm). Typically, the effective radius is less than 8 μm for very fine ash (such as in a distal plume) and between 8 and 64 μm for fine ash (Marzano et al. 2018). The range of ash optical depths is highly variable. Ventress et al. (2016) and Balis et al. (2016) recorded ash optical depths of less than 1.2 from dispersed plumes from Eyjafjallajökull in 2010; however much higher values can be expected closer to the volcano or following large explosive eruptions. The effective radius and AOD explored here for the channel selection is in the upper range and above what might be expected: values which may only be true close to the volcanic vent. The spectrum of an optically thin plume is more difficult to differentiate from a clear spectrum commonly leading to the signal ($I_{obs}(v)$-$I_{clr}(v)$) to be within the instrument noise and subsequently will result in no retrieval. A decision was made to select the channels used using idealised optically thick cases, which may only be true close to the vent, for which the plume should be evident in the majority of the $CO_2$ channels. The selected channels are tested on a wider range of AODs and effective radius in section 4.2 including smaller values that are more representative of a disperse plume.'*

**8) Section 5, page 8, line 31: "planck" should have upper-case "P".**

Done

**9) Section 5, page 9, line 33: You say that you have defined the tropopause "as the height at which the temperature profile inverts and has a positive gradient", but it's confusing that the tropopause dashed lines in Figures 6(a-f) have obviously not used this definition!**

This has been rewritten to avoid confusion. When applying the $CO_2$ slicing technique to simulated data to select the channels and then test the applicability of the technique, the procedure was allowed to return values above the tropopause to test how successfully it performed. As the method was shown to perform poorly when the temperature gradient is stable, for the application to real ash scenes, it was only allowed to retrieve up to the tropopause as defined by the WMO. Figure 6 shows the tropopause (dashed line) as defined by the WMO.

The rewritten passage:

*'Another point to note is that, in section 4, the maximum height that could be retrieved was defined as the height at which the temperature profile inverts and has a positive gradient. This is slightly above the tropopause which is defined by the World Meteorological Organisation (WMO) as the point at which the lapse rate is less than 2°C/km, and remains lower than this for at least 2 km. This was done to demonstrate how the $CO_2$ slicing method performs above the troposphere where the atmospheric temperature does not vary significantly: the atmospheric lapse rate here approaches zero. Figure 6 demonstrates that the $CO_2$ slicing method performs poorly in these cases and so in the application to real data the $CO_2$ slicing method is only allowed to retrieve values up to the tropopause as defined by the WMO.'*

**10) Section 5, page 9, lines 1-2: When you say "Figure 6 demonstrated that the $CO_2$ slicing method performs poorly where the temperature profile steepens significantly", can you clarify exactly what "steepens" means in this context – when dealing with (negative) vertical gradients, it's very easy for the reader to become confused with words such as "steepens"!**

This has been rephrased to improve clarity:

*'This was done to demonstrate how the $CO_2$ slicing method performs above the troposphere where the atmospheric temperature does not vary significantly: the atmospheric lapse rate here approaches zero.'*

**11) Section 5.2, page 10, lines 31-32: You say that "Ventress et al. (2016) identified that in some cases the retrieval assumed a lower altitude and a higher ash optical depth in order to fit the spectra". Lower/higher than what? Just needs to be a little clearer.**

We have expanded this sentence:

*'Ventress et al. (2016) identified that in some cases the retrieval underestimated the altitude of the plume and obtained a high ash optical depth in order to fit the measured spectra, when in reality the ash layer might have a lower optical depth and higher altitude.'*

**12) Figure 4: What is the x-axis here? The y-axis is labelled simply as "Wavenumber". Presumably they should both be the same as for Figure 3?**

The x axis was also wavenumber. The axes have now been labelled $CO_2$ wavenumber (x axis) and reference wavenumber (y axis).

**13) Figures 10-13: Where does the retrieved ash mass come from – the OE retrievals? In any case, it's not clear to me exactly why these mass column loadings have been included in the paper, as I don't believe they are ever referred to. What do they add to the paper?**

The ash mass can be calculated from the ash optical depth and effective radius obtained with the optimal estimation scheme (assuming an ash density). The maps of these are included for reference.

We have included an additional line within the manuscript which makes reference to these plots:

*'Some example maps of the OE heights are shown in Fig. 10 to 13b, alongside the ash mass (panel c) calculated from the OE retrievals of AOD and effective radius, assuming an ash density. The maps of ash mass show that in general the ash mass falls with transportation away from the vent.'*

As we do not refer to panel (e) of figures 10-13 within the paper we have decided to remove these.

**14) Table 3: Needs units! Presumably the "Channel Ranges" etc. refer to wavenumbers (cmˆ-1)? Are the "Peak Sensitivity Ranges" in mb/hPa?**

We've added units to the table. The channel ranges/reference channel are in wavenumber ($cm^{-1}$) and the peak sensitivity was in (hPa).